



# The stable water isotopes and snow accumulation from Weddell Sea sector imprint the large-scale atmospheric circulation variability

Andressa Marcher[1], Jefferson C. Simões[1, 2], Ronaldo T. Bernardo[1], Francisco E. Aquino[1], Isaías U. Thoen[1], Pedro T. Valente[1], Venisse Schossler[1]

[1] Centro Polar e Climático, Instituto de Geociências, Universidade Federal do Rio Grande do Sul. Av. Bento Gonçalves, 9500, CEP 91.540-000, Porto Alegre, RS, Brazil.
[2] Climate Change Institute, University of Maine, Orono, ME 04469-5790, USA.

*Correspondence to*: Andressa Marcher (andressa.marcher@gmail.com)

**Abstract.** Stable water isotopes and accumulation data extracted from polar ice/firn cores provide valuable climate information. Here, we present novel isotopic and accumulation time series from an upstream area of the Möller Ice Stream (MIS) basin, Weddell Sea Sector (WSS), Antarctica – a Brazilian research area (84°00'00''S; 79°29'39''W; 1276 m a.s.l.). Our purpose was to understand the depositional history and investigate how much the recent climate signal (21st century) is stored in the shallowest ice sheet layers in this area. Therefore, we crossed the isotopic ($\delta^{18}$O, $\delta$D, and d-excess) and snow accumulation data of two shallow firn cores (both ~9.0 m deep) with glaciological information, local and regional meteorological data (both ERA5 and AWS), indices of large-scale atmospheric modes (as SAM and ENSO) and the Amundsen Sea Low (ASL). The isotopic records cover 16 years (from 1999 to 2015-austral summer) and the accumulation records cover 20 years (from 1999 to 2018). We find that interannual $\delta$s variability is strongly explained by changes in the phase of the SAM ($r = 0.74$; $p < 0.05$; $\alpha = 0.05$) and, consequently, also by changes in pressure of both the WSS ($r = -0.57$; $p < 0.05$; $\alpha = 0.05$) and the ASL ($r = -0.56$; $p < 0.05$; $\alpha = 0.05$). The regional temperature in WSS ($r = 0.50$; $p < 0.05$; $\alpha = 0.05$) and Antarctica Peninsula ($r = 0.70$; $p < 0.05$; $\alpha = 0.05$), as well as the sea ice concentration in the Weddell Sea ($r = -0.49$; $p = 0.05$; $\alpha = 0.05$) are other factors that measurably influence the $\delta$s in the study area. In the period covered by our study, the rarest and strongest wind events (SWE; > ~15 m/s) and extreme precipitation events (EPE) oscillate almost completely out of phase, and this relationship is largely explained by the sub-decadal changes in the SWE-ENSO relationship and by the SAM variability. This oscillatory pattern between SWE and EPE justifies the non-temporally stable correlation between $\delta$s and local temperature in the study area. For the period of 2013-2018, we show that the trigger to start accumulating snow on the studied site is the occurrence of a range of EPE in a short time or of the one EPE with higher snowfall rates and that, the low snowfall events are hardly ever preserved. Our snow accumulation composite record shows that the SWE-EPE seesaw governs the snow accumulation in the upstream area of the MIS basin in the 1999-2018 period. When the frequency of SWE increases and EPE decreases, the local snow accumulation increase. In contrast, in the opposite scenario, the accumulation approaches the forecast precipitation data indicating that the influence of blowing snow and wind drift decreases. Because of this relation, incredibly there was a significant decrease in snow accumulation in the study area in the 1999-2018 period due



to an increase in EPE in recent years. Probably, in a scenario of future warming, the persistence of SAM positive trend, and the EPE increase due to intensification of wetter and warmer air masses incursions by the WSS such a relationship will change. Our results indicate that both isotopic compositions and snow accumulation are strongly influenced by large-scale modes of climate variability in the MIS basin inland. Furthermore, they also provide valuable information to understand
mass balance at the basin scale in the WSS. We recommend more shallow drills and snow pits in this site to construct the best composite record to reconstruct these atmospheric circulation patterns and solve challenges regarding the topographic effect.

## 1 Introduction

45        The Antarctic environment is a key region of the Earth's climate system because it continually exchanges water and energy with other terrestrial components (e.g., King & Turner, 2009; Turner et al., 2009; Shepherd et al., 2018; Rintoul, 2018; Holland et al., 2020), receives a range of aerosols from all over the world (e.g., Legrand & Mayewski, 1997; Arienzo et al., 2017; Delmonte et al., 2020; Marquetto et al., 2020), and integrates the carbon biogeochemical cycle (e.g., Petit et al., 1999; Siegenthaler et al., 2005; Turner et al., 2009; Rintoul, 2018). Owing to its role in the climate system, understanding the
Antarctic climate variability at different spatial and temporal scales is crucial to reducing uncertainties in long-term climate projections on global, hemispheric and regional scales and better projecting the future of Antarctica's cryosphere. However, a detailed and profound uptake of Antarctic climate variability is hampered by the spatial-temporal scarcity of modern observational and satellite records (Nicolas and Bromwich, 2014; Jones et al., 2016; Bracegirdle et al., 2019). To surpass these limitations, various proxy archives have been used to retrieve climatic information in the Antarctic environment to
highlight the ice/firn cores drilled on the ice sheet.
        Polar ice/firn cores are a vital tool to retrieve climatic information before the instrumental period, both on longer (glacial-interglacial) (Petit et al., 1999; EPICA community members, 2004; WAIS Divide Project Members, 2013) and shorter (millennial, centennial and decadal) (e.g., Küttell et al. 2012; Stenni et al., 2017; Winstrup et al., 2019; Thomas et al., 2015; Thomas et al., 2017) timescales. Particularly, shallow cores recovered in the last years provide an odd opportunity to
study the spatial climate variability on smaller time scales, as well as to calibrate the chemical signal with the instrumental record and thus improve the reconstructions of past climate (*e.g.,* Mayewski et al., 2005; Brook, 2007; Küttell et al. 2012; Altnau et al., 2015; Goursaud et al., 2019; Laepple and others, 2018; Servettaz et al., 2020; Hoffmann et al., 2020). A range of chemical and physical parameters can be measured from the ice/firn cores to reconstruct atmospheric and environmental variables (Legrand & Mayewski, 1997; Wolff et al., 2012). Nonetheless, water isotopic ratios ($\delta^{18}O$ and $\delta D$; or $\delta s$ when both
are referenced) are leading proxies of condensation temperature (Dansgaard, 1964; Masson-Delmotte et al., 2008; Jouzel, 2013; Stenni et al., 2017), and the snow accumulation rates derived from ice cores are fundamental to knowing the



local/regional surface mass balance (SMB) (Altnau et al., 2015; Fudge et al., 2016; Thomas et al., 2017; Goursaud et al., 2019; Winstrup et al., 2019; Hoffmann et al., 2020).

The potential of δs for temperature reconstructions is a result of $\delta$-depletion as the air masses cool and advance toward Antarctica inland, assumed to occur under conditions very close to equilibrium (Dansgaard, 1964; Dansgaard et al., 1973; Jouzel & Merlivat, 1984; Masson-Delmotte et al., 2008; Jouzel, 2013). Nonetheless, as the isotopic compositions imprint each process that occurs from evaporation on moisture source until deposition site and afterwards, they also store a range of other information that may either obliterate or change δs-temperature relationships on shorter timescales in a given area (Jouzel, 2013; Touzeau et al., 2016; Landais et al., 2017). For instance, the isotopic compositions of precipitation also

rely on conditions of the moisture source region: primarily sea surface temperature (SST) and relative humidity (RH) (considering the global scale), and secondarily on the presence of sea ice (in local and regional scales) (Merlivat & Jouzel, 1979; Bonne et al., 2019). Consequently, they are sensitive to variations in these conditions in the oceanic evaporative area and changes in moisture origin, linked to both fluctuations in the atmospheric circulation (Vimeux et al. 1999; Nonne & Simmonds, 2002; Nonne, 2008; Russel & McGregor, 2009; Sodemann & Stohl 2009; Naik et al., 2010) and variability of sea

ice area and concentration (Noone & Simmonds 2004; Sinclair et al., 2014; Thomas et al., 2019 and references). Since the evaporation in the source region does not occur under equilibrium conditions, d-excess (a secondary isotopic parameter sensitive to kinetic effects; calculated by: $d = \delta D - 8 \times \delta^{18}O$; Dansgaard, 1964) is used to explore changes in moisture source and atmospheric paths. This approach is especially valid in coastal and less elevated areas in the interior of Antarctica, where information about moisture sources is preserved in the air mass (Touzeau et al., 2016; Landais et al., 2017; Goursaud et al.,

2019). However, challenges regarding the interpretation of d-excess still persist in these areas due to the diversity of moisture sources (Touzeau et al., 2016; Landais et al., 2017; Goursaud et al., 2019; Marcher et al., 2022). Furthermore, some studies have shown that isotopic ratios could be biased by precipitation seasonality and intermittency (Helsen and others, 2006; Sime et al., 2008; Casado et al., 2020; Servettaz et al., 2020), as well as by extreme precipitation events (EPE; Turner et al., 2019; Servettaz et al., 2020). In addition, the isotopic content can be modified by post-depositional processes, to list:

snow mixing, erosion, and redistribution (Fisher et al., 1985, Münch et al., 2016), snow sublimation (Neumann et al., 2008), diffusion in the firn (Johnsen et al., 2000; Casado et al., 2020), and by vapour-snow exchange and re-condensation (Town et al., 2008; Steen-Larsen et al., 2014; Casado et al., 2016; Ritter et al. 2016, Bréant et al. 2019). The effects of these post-depositional processes are not decoupled and can cause noise, erase seasonal to sub-decadal (<10 years) resolution climatic information, and imprint non-climatic bias.

Annual snow accumulation derived from ice cores is computed from the depth-age relationship. In areas where the accumulation rate is high to moderate, snow layer thickness measurements obtained by the annual layer counting method can be directly applied to shallow cores to retrieve annual accumulation (*e.g.*, Lindau et al., 2016; Hoffmann et al., 2020). For deeper ice cores, because of vertical strain onto the ice sheet layers, an ice-flow model is required to adjust the annual layer thicknesses (*e.g.*, Fudge et al., 2016; Winstrup et al., 2019). It is known that the snow accumulation spatial-temporal variability is influenced by thermodynamic and dynamic mechanisms; *i.e.*, by modifications in both the relative humidity of




air masses and the atmospheric circulation on large- and synoptic-scale (Dalaiden et al., 2020a and references). In the last few decades, some studies have highlighted the importance of short-term weather events (*i.e.,* synoptic scale) to interannual and spatial variability of snow accumulation, such as EPE, linked or not to atmospheric rivers incursions, and strong wind events (SWE), linked or not to storm systems (*e.g.*, Birnbaum et al., 2010; Fujita et al., 2011; Gorodetskaya et al., 2014;

Turner et al., 2019; Dalaiden et al., 2020a; Wille et al., 2021). However, as with stable isotope compositions, there are also challenges regarding the analysis of snow accumulation on the Antarctic continent. Topographic effects can also influence the snow accumulation locally yielding non-climatic biases (Frezzotti et al., 2004; Kaspari et al., 2004). Further, although Antarctic ice cores studies have reaffirmed that positive albeit weaker coherent relations between snow accumulation and temperature are preserved in various sectors of Antarctica (*e.g.*, Dalaiden et al., 2020b), other studies have shown that snow

accumulation could not be largely related to temperature in some Holocene periods (*e.g.*, Fudge et al., 2016).

Challenges in interpreting both isotopic and accumulation records are amplified by the shortage of these records in various Antarctica sectors. Because of this and given the importance of these proxies for climatic reconstructions, shallow core studies are encouraged to know which and how much climate information can potentially be extracted from isotopic and accumulation records on a basin-scale in each Antarctica sector (Masson-Delmotte et al., 2008; Thomas et al., 2017;

Goursaud et al., 2019; Marcher et al., 2022). Here, we investigate the stable isotopic content ($\delta^{18}O$, $\delta D$, and d-excess) and the snow accumulation variability from the two high-resolution shallow firn cores drilled on the upper reaches of the Möller Ice Stream basin, Weddell Sea Sector, in the transition between West Antarctica Ice Sheet (WAIS) and East Antarctica Ice Sheet (EAIS). We aim (1) to understand the depositional history (*i.e.,* explore the annual accumulation pattern, probable events preserved, and post-depositional process), and (2) to examine which and how much recent climate information (from the late

1990s onwards) is preserved on this site. For this purpose, we reconstructed the annual snow accumulation rates. Furthermore, we investigated the precipitation intermittency and evaluated how the proxy record is constructed, analysing both the snow accumulation data from a Brazilian Automatic Weather Station (hereafter, Criosfera 1 AWS) installed in our study area and the reanalysis meteorological data (e.g., total precipitation, wind speed, and wind direction, mean sea level pressure). For comparison with the snow accumulation trend, we calculated the number of high snowfall days (HSD) and the

hourly frequency of SWE. Wind direction on high snowfall days (HSD) was separately computed in order to assess which sector comes from the storms that reach the study area. We explored the relationships between annually averaged snow accumulation and the number of HSD, frequency of SWE, the temperature at 900hPa (weighted and non-weight with precipitation), sea ice concentration (SIC) both from ABSS and WSS, large-scale atmospheric modes (as Southern Annular Mode (SAM) and El Niño-Southern Oscillation (ENSO)), and pressure in both Amundsen Sea Low (ASL) and WSS. The

same relationships were yielded for the annually averaged stable isotope composition time series to assess its climatic significance. We also considered the impact of the post-depositional processes driven by the wind and strong temperature gradients in the snowpack on isotopic compositions, as well as the influence of topography on both isotopic compositions and accumulation records. This study contributes to the understanding of the stable water isotope compositions and snow



accumulation records from the WSS inland. It also points out the potential of these records to reconstruct the large-scale
circulation variability.

## 2 Materials and Methods

### 2.1 Study area

In the present study, we use two shallow firn cores drilled at the Criosfera 1 site: the TT01 firn core (7.25 cm
diameter) and the first ~9 m of the CR1 ice core (8.50 cm diameter). Details from these cores are presented in Table 1.
Criosfera 1 site is located at MIS basin catchment area (~84°S, ~79° 30'W; at ~1276 m above sea level (a.s.l.)), near the
boundary with Foundation Ice Stream Basin (FIS basin), and approximately 650 km of South Pole (Figure 1a). These basins
lie on the Ronne Embayment region — which is drained by glaciers and ice streams that feed into the Filchner-Ronne Ice
Shelf (Figure 1a). The name Criosfera 1 is because in this area is installed the Criosfera 1 AWS (at 84° 00' 00'' S, 79° 29'
39'' W), a Brazilian AWS which is been in operation since January 2012, providing information about solar radiation, 2-m
air temperature, wind speed and direction, relative humidity, snow height, among other environmental parameters (Pinto,
2017).

**Table 1.** Details on two firn cores from the Criosfera 1 site used in this study and other information: location, date of drilling, the period
covered and depth of the cores, number of the samples used in this work, sample resolution, mean density and borehole temperature at 10
m deep.

| Cores | Coordinates / surface elevation | Drilling | Time span | Depth span | Samples | Resolution | Average density | Borehole temperature |
|-------|--------------------------------|----------|-----------|------------|---------|------------|-----------------|----------------------|
| TT01 | 83° 59' 59.5" S, 79° 29' 31.4" W; 1285 m a.s.l. | 07/01/2015 | 1999-2015 | 9.471 m | 309 | 0.03 m | 0.44 ± 0.07 g/cm$^3$ | -32 ± 0.5°C |
| CR1 | 83° 59' 59.1" S, 79° 29' 19.3" W; 1285 m a.s.l. | 02/01/2012 - 10/01/2012 | 1999-2012 | 9.130 m | 324 | 0.03 m | 0.45± 0.08 g/cm$^3$ | -32 ± 0.5°C |

In the Criosfera 1 site, the ice thickness is 1869 m and the bedrock is substantially below the mean sea level
(BEDMAP 2, Fretwell et al. 2013) (Figure 1b). The (modeled) annual snow accumulation rates are relatively moderate (10-
20 cm w. eq. y$^{-}$1; IGOS Cryosphere Theme Report, 2007; Dalaiden et al., 2020a). Currently, the basin in which the Criosfera



1 site is located (i.e., MIS basin) is stable, as well as the other basins of the Ronne Embayment. However, climatic models

point out that in a scenario of future warming and changes in heat transport in the oceans, this basin along with the Institute

Ice Stream basin would be the basins of WSS most affected by such changes (Siegert et al., 2019).

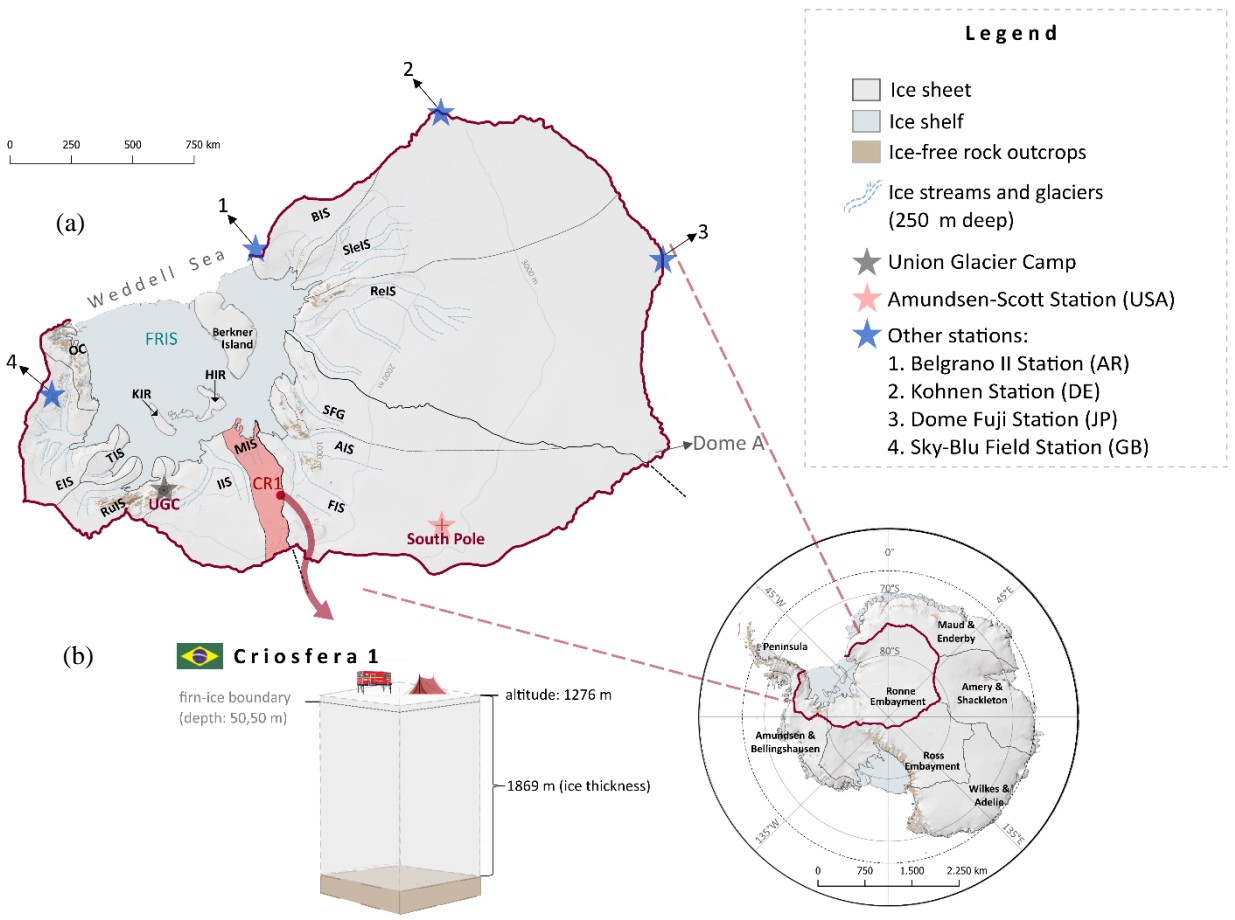

**Figure 1.** (a) Map of the Ronne Embayment basins showing the Criosfera 1 location (red dot) in the MIS basin (red area). Stars are the main stations and base camps. The thin blue dashed lines are ice streams and glacier outlines at 250 m deep. (b) A block diagram showing some geographical and general glaciology information of the Criosfera 1 site. This map was made in QGIS using the Quantarctica 3 Project datasets (Matsuoka et al. 2018). The ice thickness was extracted from the BEDMAP 2 datasets (Fretwell et al. 2013) and the depth of the firn-ice boundary was obtained from the densification model of Ligtenberg et al., 2011.


Records from Criosfera 1 AWS show that the monthly mean 2-m temperature is below zero throughout the year and the annual average temperature is ~ -31°C. Criosfera 1 site is a region characterized by strong temperature contrasts. In





summer, the temperature reaches a maximum of ~ -5°C, and in the winter, it reaches a minimum of ~ -55°C. The site experiences dry and cold air masses associated with strong katabatic winds from the Antarctic Plateau (Pinto, 2017; Marcher

et al., 2022), as a result of the configuration of the terrain in the Ronne Embayment region (Parish & Bromwich 2007). However, generally, the extremely cold condition in both late autumn, winter, and early spring is broken off by incursions of relatively warmer air masses coming from the ocean (Nicolas & Bromwich, 2011). These warm air masses can cause snowfall, increase the temperature up to -17°C, and decrease provisionally the wind speed. The prevailing wind direction is from the south (180°) and the annual average wind speed is 11.9 m/s, which indicates that the Criosfera 1 site is commonly

affected by strong wind events (SWE). Due to the extreme wind regime, the presence of sastrugi and low-scale dunes fields of 20-40 cm high in wide extension is well marked around the study area (Marcher et al., 2022).

**2.2 Firn/ice cores collection**

The CR1 ice core (106.11 m deep) was drilled in the 2011-2012-austral summer (between January 2nd and 10th) during the 29th Brazilian Antarctic Operation (Criosfera 1 AWS install campaign). A team of Brazilian researchers from the Centro Polar e Climático of the Universidade Federal do Rio Grande do Sul (CPC/UFRGS) and the National Institute for Space Research (INPE) were responsible for drilling. The CR1 core was obtained roughly 70 m east from the Criosfera 1 AWS (at 83° 59' 59.1" S, 79° 29' 19.3" W; Figure S1 in Supplementary Material) with a Fast Electromechanical Lightweight

Ice Coring System (FELICS; Ginot et al. 2002), which performs one hole drilling without the need for trench excavation.

The TT01 firn core was recovered at ~30 m east from the Criosfera 1 AWS (at 83° 59' 59.5" S, 79° 29' 31.4" W; Figure S1 in Supplementary Material), in the Brazilian Traverse to WAIS (hereafter, BR-WAIS traverse), on January 7th, 2015. BR-WAIS traverse was carried out in the 2014-2015 austral summer (between January 4th and 21st) during the 32nd Brazilian Antarctic Operation and was attended by a research group from the CPC/UFRGS. See Marcher et al. (2022) for

more details about this traverse. A Mark III auger (Kovacs Enterprises, Inc., USA) coupled with an electrical drill drive powered by a generator was used to collect the TT01 core. Unlike FELICS, it was necessary to construct a pit (2 m deep) to advance the drill to the deeper levels. The Mark III drilling system recovers one-meter long cylinder pieces. However, some pieces were retrieved at uneven lengths and broke up into smaller pieces due to broad changes in snow hardness in the study area.

The pieces of both cores were weighed on a lab scale (Ohaus®; precision: 0,01g), and then densities were determined using the core diameter, length, and weight measurements. In addition, an accurate description of the structures and properties of firn/ice was carried out for stratigraphic studies. All pieces were packed in a polyethylene bag and stored in the high-density Styrofoam box by protocol. From Antarctica, the cores were transported frozen two times by air. The first air transfer was to Punta Arenas (Chile), and the second was to New York (USA). After, the material was sent by road to the



Climate Change Institute (CCI), University of Maine (UMaine; USA), where it was decontaminated and subsampled in
certified class 5 cold and clean rooms (ISO 14644-1, 1999).

## 2.3 Procedures performed in certified cold and clean room

### 2.3.1 Decontamination

Both cores were decontaminated in a class 5 cold room at a temperature below -20°C using the method of Tao et al.
(2001). A brief description of the methodology used is given below. Thin layers on the sides (4 mm thick) and ends (2 mm
thick) of each piece of firn were removed using a pre-cleaned ceramic (ZrO) knife under a clean bench with a laminar flow
system. Visual description of the stratigraphy was then enhanced. Pieces were conditioned in pre-cleaned acrylic tubes
before being moved to another clean room for melting. All labware used was carefully pre-cleaned with acid baths and
rinsed with type I water from the Milli-Q system (resistivity > 18 MΩ cm). During the decontamination step (and also during
melting and subsampling; see next subsection 2.2.2.), Tyvek® protective suits (DuPont™, Wilmington, DE, USA) and
polyethylene (PE) gloves were worn to reduce contamination.


### 2.3.2 Subsampling

The cores were melted using a continuous ice melting system with discrete and high-resolution sampling (CMDS)
developed by CCI researchers (Osterberg et al., 2006). This system is installed in class 5 clean room with high efficiency
particulate air (HEPA) filters. It consists of a refrigerator equipped with a modified Wagenbach-style Ni270 controlled
heating plate and coupled with automatic fraction collection systems (Gilson Company, Inc.). The Ni melting plate contains
a series of radially arranged micrometre (µm) slits and a circular separator ridge that allows separation of the melt water
originating from the edges of the core from that originating from the central part of the core (Osterberg et al., 2006). As the
melting happens, the melt water is diverted through channels to a clean bench equipped with three Gilson FC204 automatic
fraction collectors by peristaltic pumping. The outer channels direct the water for the stable isotope sample collector, while
the inner channel directs the water for the other two collectors located in a laminar flow air hood for sample collection
intended for the analysis of ion content and trace elements. The melting temperature is manually controlled according to the
density of each firn cylinder. It was set to about 17.9±1.4°C for TT01 and 16.3±1.0°C for melting the first ~9 m of CR1.
Samples for isotopic analysis were stored in 25-ml vials of high-density polyethylene (HDPE), whereas samples for ion
chemical analysis were stored in 5-ml vials of polypropylene (PP). Before use, the PP vials were triple rinsed in type I water,



soaked in type I water for 24 hours, triple rinsed again in type I water, and dried under a class 5 clean bench as recommended by Curran & Palmer (2001) and Osterberg and co-authors (2006). The CMDS system is always washed with ultrapure water before and after the melting step, and water blanks are collected.

Ice samples from both cores remained stored in a cold storage facility in Maine (USA) until 2018 when they were
sent to CPC/UFRGS (Brazil). There, they were kept frozen until analysis.

## 2.4 Measurements

The refrozen samples of the two cores were analysed at the laboratories of the CPC/UFRGS (Brazil). Table 2
summarizes the analytical method performed in each core, the chemical parameters, and the number of samples analysed.

**Table 2.** Techniques applied and chemical parameters analysed in each core (TT01 and CR1). The number of samples is also listed. In bold are trace ionic species used in this study.

| Cores | Analytical technique | Parameter analyzed | Number of Samples |
|-------|----------------------|--------------------|-------------------|
| TT01 | WS-CRDS | $\delta$D and $\delta^{18}$O | 309 |
|       | IEC | Ionic content: **Na$^+$**, NH$_4^+$, K$^+$, Mg$^{2+}$, Ca$^{2+}$, **Cl$^-$**, NO$_3^-$, **SO$_4^{2-}$**, C$_2$O$_4^{(2-)}$ | 309 |
| CR1 | WS-CRDS | $\delta$D and $\delta^{18}$O | 324 |

### 2.4.1 Snow isotopes analysis

The stable water isotope analysis was performed at the Stable Isotope Lab using two wavelength-scanned cavity ring-down spectroscopy (WS-CRDS) systems (PICARRO® L2130-i, USA) — one coupled with a Combi PAL Autosampler (CTC Analytics AG, Switzerland) and the other coupled to a Picarro Autosampler (PICARRO® A0325, USA). All hydrogen
and oxygen isotopes ratios results are represented as δ-notation — *i.e.*, deviation per mille (‰) of the sample isotopic ratios (R = D/H or $^{18}$O/$^{16}$O) in relation to the V-SMOW (Vienna Standard Mean Ocean Water) reference standard ratios, according to the Eq.1:

$$\delta_{(sample)} = \left( \frac{R_{(sample)}}{R_{(VSMOW)}} - 1 \right) \times 1000 (‰) \qquad \text{(Eq. 1)}$$


For a detailed description of our laboratory routine, see Oliveira 2019 and Marcher et al. (2022). Briefly, to correct for memory effect and instrumental drift, eight injections were performed for each sample and the first five results were


discarded. Then, the average of the last three injections was taken. Finally, the mean raw stable isotope ratio datasets were calibrated with linear regression curves constructed using three secondary water laboratory standards. Our laboratory

standards were previously normalized to the international VSMOW2 - GISP (Greenland Ice Sheet Precipitation) - SLAP2 (Standard Light Antarctic Precipitation) scale.

The d-excess value was calculated using the linear definition (Dansgaard, 1964): $d = \delta D - 8 \times \delta^{18}O$. The accuracy measurement was better than 0.9‰ and 0.2‰ for $\delta D$ and $\delta^{18}O$, respectively. This resulted in an error of ~1.7‰ for d-excess.

**2.4.2 Ionic chemistry analysis**

Ionic chemistry analysis was conducted at the Glaciochemistry Lab – a clean lab classified as class 7 (ISO 14644-1, 1999). We used two Dionex™ ion-exchange chromatography (IEC) systems with a modified sample loop volume (250 µl), eluent autogenerator, electro-chemical suppression, and conductivity detectors (Thermo Fisher Scientific Inc., USA). The

trace ion species measured in this study were sodium ($Na^+$), ammonium ($NH_4^+$), magnesium ($Mg^{2+}$), calcium ($Ca^{2+}$), chloride ($Cl^-$), nitrate ($NO_3^-$), sulphate ($SO_4^{2-}$), and oxalate ($C_2O_4^{(2-)}$) (as listed in Table 2). A Dionex™ ICS-2100 equipped with Ion Pac™ CS12A (2 mm) analytical and Ion Pac™ CG12A (2 mm) guard columns was used for cation analysis; while a Dionex™ ICS-2000 equipped with Ion Pac™ AS17C (2 mm) analytical and Ion Pac™ AG17C (2 mm) guard columns was used for anions analysis. The use of two analytical pieces of equipment and a flow separator in the path that connects the autosampler

to the analytical systems allowed the simultaneous analysis of cations and anions.

The analytical routine applied was a slight adaptation of the methodology of Thoen et al. (2018). Briefly, we altered the eluent generation gradients and thus increased the analysis time of each sample (to 26 min) in order to improve analytical performance. All samples and standard solutions were prepared in a class 5 environment inside the vertical laminar flow air hood before each analysis round. The 5 ml PP vials suitable for autosampler (Dionex™ Polyvials) were precleaned with

ultrapure water (conductivity of 0.054 µS/cm) according to the aforementioned protocol (Curran & Palmer; 2001; Osterberg et al., 2006). Diluted standard solutions were prepared from pure salts (for analytical datasets calibration and equipment checking) and from the stocks of commercial Standard Reference Materials (SRMs; for equipment checking) using ultrapure water. The standard solutions were designed to contain a mixture of all ions of interest in concentrations similar to levels expected in a polar ice matrix (in ppb level).

All trace ions were expressed in concentrations of microgram per litre ($\mu g\ L^{-1}$). For quality control, the composition of the ultrapure water yielded by the laboratory's Milli-Q system was routinely verified. Blanks from the melting step were also analysed, and their mean results were calculated for subtraction from the analytical datasets (according to Thoen et al., 2018). In this paper, we use only $Na^+$, $Cl^-$, and $SO_4^{2-}$ (also non-sea-salt (nss) $SO_4^{2-}$, which is introduced in the next section) datasets.




## 2.5 Dating of firn cores and estimates of annual snow accumulation rates

TT01 firn core was dated by the manual annual layer counting (ALC) method. Seasonal variations of $nssSO_4^{2-}/Na^+$ and $Cl^-/Na^+$ ratios (both peaking in summer), as well as the ratios of water isotopes ($\delta$s) and individual chemical species

(such as $Na^+$ (peaking in winter), $SO_4^{2-}$, and $nssSO_4^{2-}$ (peaking in summer)), were used as annual-layer markers. We assumed that $Na^+$ was essentially derived from sea salt (ss) and estimated the $nssSO_4^{2-}$ using the well-known formula (Eq. 2):

$$[nssSO_4^{2-}] = [SO_4^{2-}] - 0.251 \times [ssNa^+] \tag{Eq.2}$$

where 0.251 is the $[SO_4^{2-}/Na^+]$ weight ratio in seawater and $SO_4^{2-}$ and $ssNa^+$ are the concentrations of these ions in snow/firn samples. The $nssSO_4^{2-}$ refers to the sulphate fraction that is not derived from sea spray. In Antarctica, the $nssSO_4^{2-}$ aerosol comes primarily from atmospheric oxidation of dimethylsulfide (DMS), a substance released from phytoplankton blooms that typically occur from late spring to late summer. However, it can also come from other sources, such as polar and extrapolar volcanic activity, pollution, and others (Delmas, 1994; Udisti, 1996; Legrand & Mayewski, 1997; Steig et al.,

2005; Abram et al., 2013). Regardless of the source, maximum concentrations of this sulphate aerosol in Antarctic snow are expected to occur in early or mid-summer as the polar vortex breaks up in springtime (Steig et al., 2005). $Na^+$ can be considered a proxy for marine air mass advection (Mayewski et al., 2017; Raphael et al., 2016), and its highest concentrations in wintertime are due to stronger atmospheric circulation strength in this period (Legrand & Mayewski, 1997; Sigl et al., 2016; Mayewski et al., 2017).

Here we chose the $nssSO_4^{2-}/Na^+$ ratio as the primary marker of the time (i.e., peaks indicate New Year's Day) instead of other parameters because it has a more defined seasonality. This ratio has been widely used as a primary time marker in Antarctica (e.g., Sigl et al., 2016), as has the nssS/Na ratio obtained by ICP-MS analysis (e.g., Sigl et al., 2016; Arienzo et al., 2017; Hoffmann et al., 2020; Marquetto et al., 2020). In order to increase the accuracy of ACL dating, we used the $nssSO_4^{2-}/Na^+$ ratio along with the $nssSO_4^{2-}$ data and global volcanism historical records (available at:

https://volcano.si.edu/; last access: May 2022) to identify explosive volcanic events and compared the $\delta D$ and $nssSO_4^{2-}$ trend from TT01 with that from another firn core drilled 7.5 km west of CR1 AWS (Lindau et al., 2016). We also seek support in other cores from the WSS.

Unfortunately, we used only the seasonal variability of the $\delta$s for dating the first ~9 m of the CR1 ice core because ionic chemistry analysis has not yet been performed. Nevertheless, we tried improving the dating of CR1 by synchronizing it

with the TT01 record using water isotope ratios. However, due to the noisier signal from the CR1 core, this approach was very challenging.

After dating both cores, annual snow accumulation rates were determined and presented in meters of water equivalent per year (m w. eq. $y^{-1}$). For this, the real depth was previously converted in m w. eq. using the densities of the core sections calculated during the fieldwork (section 2.1).



## 2.6 Meteorological data


In this study, we use both reanalysis and AWS data. CR1 AWS datasets (surface temperature, snow accumulation, wind speed, and wind direction) were downloaded from the Criosfera 1 website (available at: https://www.criosfera1.com/criosfera-database; last access: January 2022). Because the CR1 AWS was installed in January
2012, it covers less than a quarter of the time span of TT01 and does not cover the period of the CR1 core. Therefore, we used the CR1 AWS data only to validate the performance of the reanalysis model solutions, to assess the main meteorological information, and to understand the accumulation scenarios at the Criosfera 1 site.

We chose to use the European Centre for Medium-Range Weather Forecasts (ECMWF) Reanalysis 5 (ERA5) datasets for comparison with isotopic and accumulation records. ERA5 is the fifth generation of ECMWF reanalysis and the
latest version released of all reanalysis models. Compared to other reanalysis products, it has higher spatial (with a lat-long grid of 0.25° and 37 pressure levels) and temporal (from 1950 to present) resolution and better global precipitation and evaporation balances (Hersbach & Dee 2016, Hersbach et al. 2019). In addition, the ERA5 satisfactorily reproduces the variability of the meteorological parameters at the Criosfera 1 site, although overestimates the temperature and underestimates the wind velocity (see Table S1). ERA5 datasets were obtained from the Climate Data Store hosted by
Copernicus (CDS (C3S) 2017; available at: https://cds.climate.copernicus.eu/cdsapp#!/home; last access: May 2022). Table 3 list the ERA 5 datasets and meteorological variables used in our study. We extracted the hourly ERA5 reanalysis variables from the CR1 AWS grid point (-84 (lat); -79.5 (long)) using an adapted version of the methodology of Hufkens et al. (2019). Monthly sea ice time series were obtained for ABSS (grid area: 120°W-60°W; 60°S-90°S) and WSS (grid area: 60°W-20°E; 60°S-90°S) through the Climate Reanalyzer.org (CCI/UMAINE; available at:
https://climatereanalyzer.org/reanalysis/monthly_tseries/; last access: March 2022). Further, monthly mean sea level pressure data from WSS were also extracted through Climate Reanalyzer.org.

**Table 3.** Variables extracted from the ERA5 datasets.

| Products | Datasets | Variables | Period |
|----------|----------|-----------|--------|
| | reanalysis-era5-single-levels | 2m_temperature | 1998-2018 |
| | reanalysis-era5-single-levels | total_precipitation | 1998-2018 |
| ERA5 hourly data | reanalysis-era5-single-levels | 10m_u_component_of_wind | 1998-2018 |
| from 1979 to present | reanalysis-era5-single-levels | 10m_v_component_of_wind | 1998-2018 |
| | reanalysis-era5-pressure-levels | temperature (900 hPa) | 1998-2018 |
| | reanalysis-era5-pressure-levels | u_component_of_wind (850 hPa) | 1998-2018 |
| | reanalysis-era5-pressure-levels | v_component_of_wind (850 hPa) | 1998-2018 |
| ERA5 monthly data | reanalysis-era5-single-levels | sea_ice_concentration | 1998-2018 |
| from 1950 to present | reanalysis-era5-single-levels | mean_sea_level_pressure | 1998-2018 |




The time series of the large-scale atmospheric modes (SAM and ENSO) and the geographical position and strength of the low-pressure climatological zone from the Pacific sector (ASL) were also compared with the time series of the water stable isotope compositions and accumulation rates. Marshall SAM index (Marshall, 2003; available at: https://legacy.bas.ac.uk/met/gjma/sam.html; last access: January 2022) was used to quantify the SAM phase, while the
Southern Oscillation index (hereafter SOI index; NOAA; available at: https://www.ncdc.noaa.gov/teleconnections/enso/soi last access: January 2022) was used to quantify the ENSO phase. We used the ASL indices (Hosking et al., 2016; available at: http://scotthosking.com/asl_index; last access: January 2022) to describe the geographical position and strength of the ASL.

## 2.7 Computation of daily snow height, high snowfall days and strong wind events

To understand the history of snow accumulation at the Criosfera 1 site we compared the snow height and wind speed records from the Criosfera 1 AWS with reanalysis precipitation data. The noise of snow height data was minimized by calculating the daily accumulation. We also evaluated the high snowfall days (hereafter, HSDs) associated with extreme
precipitation events (EPEs) and the relative hourly frequency of strong wind events (hereafter, SWEs) to comprise the accumulation at the studied site. HSDs and EPEs were computed according to the methodology of Welker et al. (2014). The calculated HSD threshold for the Criosfera 1 site in the period 1998-2018 is 2.43 mm/day. Here, in this study, we considered SWEs as those events with wind speeds above 15.51 m/s. Such threshold corresponds to the 95$^{th}$ percentile of all hourly wind speeds from 1998 to 2018. This approach was adopted because the average wind speed at the Criosfera 1 site (~12 m/s)
is slightly higher than the frequent threshold adopted for SWEs (10 m/s; Yu and Zhong, 2019 and references), therefore we only consider the strongest and rarest SWE. Our threshold for SWEs is below of that considered in coastal areas: from gale velocities (> 17.2 m/s), according to the Beaufort Wind Force scale (Turner et al., 2009b). Since the coastal areas are windier than inland, we have assessed our SWEs threshold as appropriate for intermediate areas of the continent. In addition, we analysed the wind direction at 850 hPa in the HDS to verify from which direction the EPEs that reach the studied site come.

## 2.8 Time series construction and trend and correlation analysis

We constructed composite records of the annual average of each isotopic parameter and of annual snow accumulation for the Criosfera 1 site. For this, before taking the averages between the different records, we standardized the
isotopic time series extracted from each core (CR1 and TT01 cores) and the accumulation series obtained from both the cores and the Criosfera 1 AWS using the local mean and standard deviation. We evaluated trends for the studied period and performed correlations using both composite and standardized records.





Annual averages of precipitation, SIC in ABSS and WSS, mean pressure at sea level in the Weddell sector, and the temperature at 900hPa unweighted and weighted with precipitation (and also with HSD) were computed for the construction

of the time series. The 900hPa level was chosen because it is near the cloud baseline and because it is above the inversion layer. Annual temperature weighted with precipitation (and HSD) was calculated using Equation S1 (Supplementary Material). In order to evaluate the climatic value preserved in the Criosfera 1 site, we apply the approach described below. Correlations among all these series of climatic parameters previously mentioned, the number of HSD and relative hourly frequency of SWE per year, the series of SAM, SOI, and ASL indices, and the annual records of snow accumulation were

performed. The same climatic indices and parameters were also correlated with firn core isotopic records. The series of precipitation, temperature, SIC, mean pressure at the level, number of HSD, and frequency of SWE used in the correlations were detrended. Five year running correlations among isotopic and accumulation signals, climatic parameters, and indices were also carried out to assess the behaviour of the relationships between these parameters over time.

**3 Results**

**3.1 Firn core age model**

The results of core dating were previously presented in Table 1. Figure 2 shows the age-scale constructed for TT01

shallow firn core by ALC. In general, ionic content proved better seasonal markers than isotopic content due to precipitation intermittency and diffusion already observed in the first ten meters of the snowpack in the Criosfera 1 site.

The depth-age scale of TT01 is based mainly on the $nssSO_4^{2-}/Na^+$ ratios due to its clear seasonal cycle. To improve the accuracy of our dating, we investigated the extreme peaks of $nssSO_4^{2-}/Na^+$ and $nssSO_4^{2-}$ data in order to identify explosive volcanic events. We identified a peak of $nssSO_4^{2-}/Na^+$ more than 3σ above the average that we associated with

Puyehue-Córdon Caulle Chilean volcanism (VEI = 5; a stratovolcano located at 40.59 °S and 72.12 °W, Chile) (Koffman et al., 2017). This volcanic event occurred in June 2011, but some studies identified the chemical signature of this event in the snowpack in early 2012 (e.g., Hoffmann et al., 2021). We suspect that the significant peak identified in 2008 (>3σ) is associated with other two Chilean explosive volcanic events: the emissions of Chaiten (May 2, 2008; VEI = 4; a caldera located in 42.83°S and 72.65 °W) and Llaima (January 1, 2008; VEI = 3; a stratovolcano located in 38.69°S and 71.73°W)

volcanism. Nonetheless, a study carried out in O'Higgins Antarctic Chilean base (in Antarctica Peninsula) showed that the Chaiten's chemical signal was detected in the snow from August 2008, four months after the primary eruption (Cid-Agüero et al., 2017). As our signal has no such delay, more studies are necessary to confirm the signature of these volcanisms and to verify if there was rapid tropospheric transport to the Criosfera 1 site. We also compared our chemical signal with other ice cores drilled in the WSS. It was detected correlated peaks of $nssSx/Na^+$ in 2006 between TT01 and PASO-1 firn core (drilled

at 79°38'00.68''S, 85°00'22.51''W; near the Ellsworth Mountains; Hoffmann et al. 2020). In addition, we realize that both





δD and nssSO$_4^{2-}$ time series of the TT01 and BR-IC-4 (drilled at 83°58'59.4''S and 80°07'01.4'' W; Lindau et al., 2016) firn cores interrelate well in its overlapping period (1999-2003). This well match is more marked in 1999-2000, where both cores showed prominent nssSO$_4^{2-}$ peaks of the same magnitude (>100 µg L$^{-1}$) and a δD maximum peak of ~ -255‰ in summer 2000. Both volcanic events identified and matched between cores were used as absolute time horizons during the timescale

elaboration. In total, the TT01 firn core cover 16 years (from 1999-2015), with an estimated uncertainty of ± 0.41 (< 5 months).

The dating of the CR1 ice core is presented in Figure S2 in Supplementary Material. The depth-age scale of CR1 is based mainly on the δs seasonal cycle. To refine the CR1 dating, we compared their isotopic trends with those of TT01 firn core. We verify that the isotopic trends roughly resemble only in 2011 and the period 2003-1999. In summary, the first 9 m

of the CR1 core has a shorter record than TT01 (13 years; from 1999 to 2012) and the estimated uncertainty of CR1 dating is ± 0.53 (~6 months).





**Figure 2.** Dating of the TT01 firn core based on the counting and inter-matching of maximum peaks of (a) stable isotope ratios ($\delta$D), (b) nssSO$_4^{2-}$/Na$^+$, (c) Cl$^-$/Na$^+$, (d) nssSO$_4^{2-}$ and minimum of Na$^+$ (e). All time series were smoothed every 3 points by running average. Grey dashed lines indicate the years. Coloured vertical lines indicate correspondence with other firn cores drilled in the WSS: with the nssS/ssNa data of the PASO-1 firn core (green line; Hoffman et al., 2020) and the $\delta$D and nssSO$_4^{2-}$ data of the BR-IC-4 firn core (red lines; Lindau et al., 2016). Identified Chilean explosive volcanism events (confirmed or suspected) are highlighted in faded red. Depth is presented in meters of water equivalent (w. eq.).



## 3.2 Stable water isotopes results and glaciological information from Criosfera 1 site

Basic statistics on the isotopic composition of each core are summarized in Table 4, and the isotopic, density, and stratigraphic profiles are presented in Figure 3. The stable isotope compositions of the two firn cores are comparable in mean, standard deviation, minimum, and maximum values (Table 4). However, the minimum and maximum of the cores mismatch over time (Figure 3). The isotopic signal pattern of both cores clearly indicates that not all precipitation events are preserved on the Criosfera 1 site. Furthermore, it shows a greater tendency to store relatively warmer events. Remarkably, the CR1 has a signal nosier than TT01. Even though the signal has been smoothed every three points by running average, the

presence of a saw tooth signal superimposed on the principal CR1 isotopic signal is still observed (Figure 3). In addition, as previously mentioned, the CR1 core covers a shorter period than the TT01 core, although they have the same depth span.

**Table 4.** Mean (and standard deviation), minimum and maximum values of the isotopic content ($\delta^{18}$O, $\delta$D, and d) for each core collected at the Criosfera 1 site.

| Cores | | $\delta^{18}$O (‰) | $\delta$D (‰) | d-excess (‰) |
|---|---|---|---|---|
| | Mean | -39.10 | -308.32 | 4.5 |
| TT01 | SD | 2.57 | 21.20 | 2.2 |
| (2015-1999) | Min | -45.24 | -355.67 | -0.8 |
| | Max | -32.60 | -256.02 | 11.5 |
| | Mean | -38.88 | -307.71 | 3.4 |
| CR1 | SD | 2.19 | 17.78 | 1.8 |
| (2012-2000) | Min | -43.57 | -343.93 | -0.9 |
| | Max | -32.65 | -256.22 | 8.1 |


The core density ranged from 0.19 to 0.53 g cm$^{-3}$ and from 0.23 to 0.57 g cm$^{-3}$ for TT01 and CR1 cores, respectively (Figure 3). As shown in Table 1, the average density for both cores is practically the same. An average density profile for the Criosfera 1 site can be seen in Figure S3 (Supplementary Material). To construct this average density profile, we resampled the densities of the CR1 core to the TT01 sampling scale then we took the average densities between cores. On

average, the densities in the Criosfera 1 site ranged from 0.23 to 0.54 g cm$^{-3}$ and the first two meters have an average density of 0.37 g cm$^{-3}$ (Figure S3).

The main stratigraphic features observed at Criosfera 1 site are summarized in the following. Depth hoars — *i.e.,* large-grained and faceted crystals — were observed in both cores. Particularly, they occur between the maximum and minimum peaks of δs. We also identified 14 very thin (< 5 mm) and discontinuous ice lenses in the TT01 firn core: where

nine ice lenses lie near the δs maximum peaks and five near the δs minimum (Figure 3). Unfortunately, the ice lenses were not described in the CR1 core.





### 3.3 Co-isotopic relationships

Figure 4 compares our study's δD-δ$^{18}$O relationships with the global meteoric water line (GMWL: δD = 8 × δ$^{18}$O + 10; defined by Craig 1961) and the Antarctic Ice Sheet line (AISL: δD = 7.75 × δ$^{18}$O – 4.93; Masson- Delmotte et al. 2008). Our Local Meteoric/Ice Sheet line has both the slope (8.13 ± 0.03‰ ‰$^{-1}$) and the intercept (9.11 ± 1.34‰) very close to that of the GMWL. This proximity is important because it ensures the reliability of our data and may indicate that the isotopic signal is maintained throughout the moisture transport in the atmosphere and the snowpack of the Criosfera 1 site (Craig,

1961; Dansgaard, 1964; Masson-Delmotte et al., 2008).

We evaluated the co-isotopic relationships between d-excess and δs (Figures 4d and 4e), but no linear relation was found (Table S2; Figure 6). Nonetheless, it was observed in both cores that the trends of the d-excess/δs slopes and d-excess-δs correlations time series are coherent and have an oscillatory trend (Figures 4d and 4e). In addition, the co-isotopic trends of the TT01 core roughly temporally match with ones of the CR1 core. Changes in these two relations are unjustified by

stratigraphic characteristics (e.g., ice lenses and depth hoar).

### 3.4 Isotopic trends and correlations with climatic parameters and indices

The trends of the isotope compositions time series of both firn cores (TT01 and CR1) were analysed. TT01 firn core

presented non-significant isotopic trends from 1999 to 2014. The δ$^{18}$O$_{TT01}$ and δD$_{TT01}$ trends are decreasing, with slope of - 0.01 ‰ yr$^{-1}$ ($p > 0.05$ (0.75); $\alpha = 0.05$) and -0.01 ‰ yr$^{-1}$ ($p > 0.05$ (0.87); $\alpha = 0.05$), respectively. The d-excess$_{TT01}$ is increasing, with slope of 0.05 ‰ yr$^{-1}$ ($p > 0.05$ (0.20); $\alpha = 0.05$). The CR1 core also presented non-significant isotopic trends, but for the period 2000-2011. Both δ$^{18}$O$_{CR1}$, δD$_{CR1}$ and d-excess$_{CR1}$ trend is increasing, with slope of 0.05 ‰ yr$^{-1}$ ($p > 0.05$ (0.33); $\alpha = 0.05$), 0.05 ‰ yr$^{-1}$ ($p > 0.05$ (0.26); $\alpha = 0.05$) and 0.06 ‰ yr$^{-1}$ ($p > 0.05$ (0.14); $\alpha = 0.05$), respectively. For

the same period of CR1, the TT01 trend is also increasing, with δ$^{18}$O$_{TT01}$ slope of 0.04 ‰ yr$^{-1}$ ($p > 0.05$ (0.56); $\alpha = 0.05$), δD$_{TT01}$ slope of 0.05 ‰ yr$^{-1}$ ($p > 0.05$ (0.46); $\alpha = 0.05$), and d-excess$_{TT01}$ slope of 0.11 ‰ yr$^{-1}$ ($p > 0.05$ (0.06); $\alpha = 0.05$).





**Figure 3.** Stable water isotopes (δ¹⁸O, δD, and d-excess), density, and stratigraphic profiles for (a) TT01 firn core and (b) CR1 ice core. All isotopic time series were smoothed every 3 points by running average.





**Figure 4.** Criosfera 1 co-isotopic relationships. (**a**) Relationship between $\delta^{18}O$ and $\delta D$ for Criosfera 1 site computed with TT01 and CR1 isotopic datasets. (**b**) show the same relationship but only for the TT01 isotopic dataset. (**c**) same as (**b**) but for the CR1 isotopic dataset. Grey dots are data points of the firn cores. (**d**) Temporal variability of the relationship between $\delta^{18}O$ and d-excess for TT01 firn core: a plot of ten points running slope (‰ ‰$^{-1}$; pink line) and running correlation coefficient (r; blue line). The polynomial fits represent the trend of running slopes (pink dotted line) and running correlations (blue dotted line). (**e**) same as (**d**) but for the CR1 core.

Composite records were constructed for each isotopic parameter in order to better evaluate the isotopic trends in Criosfera 1 site (Figure 5). As shown in Figure 5, these records cover the period from 2000 to 2011. For this period, none




statistically significant isotopic trend was observed to δs: the $\delta^{18}O_{composite\ record}$ slope is 0.05 ‰ yr$^{-1}$ (p > 0.05 (0.26); α = 0.05) and the $\delta D_{composite\ record}$ slope is 0.05 ‰ yr$^{-1}$ (p > 0.05 (0.18); α = 0.05). On the other hand, a statistically significant trend is verified in d-excess$_{composite\ record}$ (slope of 0.09‰ yr$^{-1}$ (p < 0.05 (0.04); α = 0.05)).


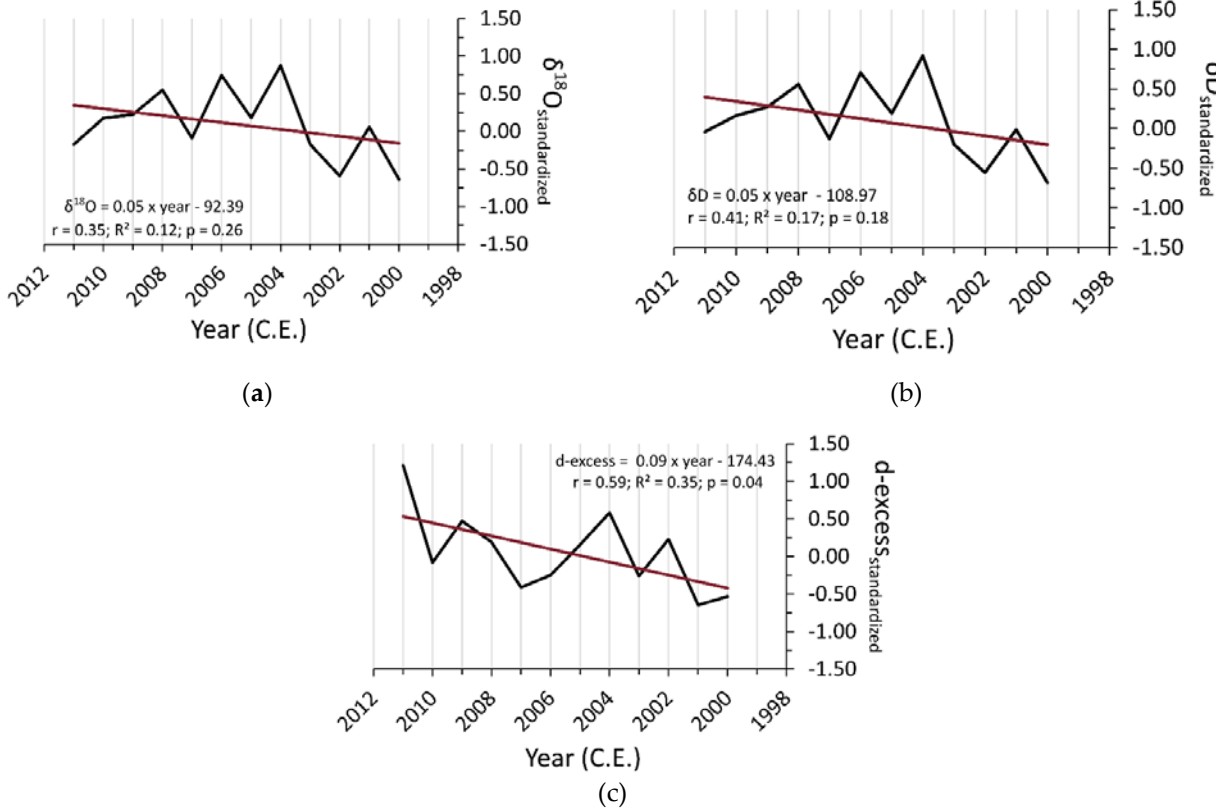

**Figure 5.** Mean annual of stable water isotopes compositions at Criosfera 1 site for the period from 2000 to 2011. Composite records of (a) $\delta^{18}O$, (b) $\delta D$, and (c) d-excess computed from standardized data are shown. Linear trend lines are indicated in red. Equation, correlation, determination coefficients, and p-values of linear regression are given for each isotopic parameter.


We computed linear correlations on the annual scale among (standardized) stable water isotope compositions, climatic parameters, and indices in a 95 percent confidence interval (α = 0.05). Figure 6 shows these correlations for the TT01 time span (1999-2014), while Table S2 (Supplementary Material) lists these correlations for the CR1 and composite

record time span (2000-2011). Five year running correlations were yielded to verify the stability of some of these correlations over time (see Figure S4 in Supplementary Material). Here, we mainly focus on the relationships obtained for



the TT01 core in order to evaluate the isotopic-climatic relations at Criosfera 1 site for the longest possible term. Further, we evaluated that the isotopic records of the TT01 would be more appropriate to perform climatic correlations due to the nosier signal of the CR1 core (section 3.2) and because of the composite record mirror relations obtained for the TT01 core (Table S2).


The $\delta s_{TT01}$ had a significant, strong positive correlation with the SAM index (r = 0.74; p < 0.05) (Marshall, 2003), and a significant, moderate negative correlation with the mean pressure in the WSS (r = -0.57; p < 0.05) and central pressure of the ASL (r = -0.56; p < 0.05). Positive correlations between (non-standardized) annual $\delta s_{TT01}$ and seasonal SAM indices (not shown in Figure 6) are also observed: significant and strong correlations with winter (r = 0.70; p < 0.05; α = 0.05) and spring (r = 0.68; p < 0.05; α = 0.05) SAM indices and non-significant weak correlation with autumn (r = 0.35; p > 0.05; α = 0.05) and summer (r = 0.11; p > 0.05; α = 0.05) SAM indices. Running correlations show that these $\delta s_{TT01}$-$SAM_{index}$, $\delta s_{TT01}$-$WSS_{sector\ pressure}$, and $\delta s_{TT01}$-$ASL_{central\ pressure}$ relations were relatively stable from 1999 to 2014 (Figures S4a and S4b; Supplementary Material), with the exception of the $\delta s_{TT01}$-autumn and $\delta s_{TT01}$-summer SAM indices relationships (Figure S4a).



It is also worth mentioning the observed relations between δs and SIC and ASL zonal position. We found moderate correlations stable over the time between $\delta s_{TT01}$ and $WSS_{SIC}$ (negative; r = -0.49; p = 0.05; Figure 6; Figure S4c) and no linear correlation between $\delta s_{TT01}$ and $ABSS_{SIC}$ (Figure 6). The lack of correlation between $\delta s_{TT01}$ and $ABSS_{SIC}$ is a result of the change in the relationship between these two parameters that occurred from 2005 onwards (Figure S4c). The correlation between $\delta s_{TT01}$ and $ASL_{longitude}$ was moderate, positive (r = 0.49; p =0.05), and stable from 1999 to 2008, but it changed to negative in 2009 (Figure S4d).


No statistically significant relationship between δs and the local temperature was verified on the annual scale in the period studied (1999-2014) (Figure 6; Table S2). Running correlation between standardized $\delta^{18}O_{TT01}$ (and $\delta^{18}O_{CR1}$) and the annual average temperature at 900hPa (weighted with precipitation; detrend) are shown in Figure S5. The δs-temperature relationships were not stable over a short period (< two decades) at Criosfera 1 site: it was positive in 2009-2014 and 1999-2002 and tended to be negative in the 2003-2008 period. However, we observed that standardized δs correlate positively and moderately with regional mean 2 m temperature (detrended) in WSS ($\delta^{18}O_{TT01}$-2 m temperature $_{WSS}$ relationship: r = 0.50; p = 0.05; α = 0.05), considering the area between the longitude of -60°W and 0°, and the latitude of -60°S and -80°S. In addition, stronger positive correlations between standardized $\delta^{18}O_{TT01}$ and regional mean 2 m temperature (non-detrended) in the western area of WSS (moderate to strong correlation) and Antarctic Peninsula region (strong correlation) were also verified (Figure S6).



The d-excess correlated moderately with the latitude of ASL (significant and positive correlation; r = 0.5; p < 0.05) and weakly with both $ABSS_{SIC}$ (non-significant and positive correlation; r = 0.29; p > 0.05) and $WSS_{SIC}$ (non-significant and negative correlation; r = -0.38; p > 0.05) (Figure 6). The d-excess$_{TT01}$-$ASL_{latitude}$ relationship was positive over the 2002-2014 period (Figure S4e) and stable from 2005 onward. While the positive d-excess$_{TT01}$-$ABSS_{sic}$ and negative d-excess$_{TT01}$-$WSS_{sic}$





relationships were established from 2000 (Figure 6), being that the former relation was stable from 2002 to 2014 and the

latter was not stable over the 1999-2014 period (Figure S4f).

To verify if interannual variations in the frequency of EPE and SWE explain the stable water isotope compositions

variability at the Criosfera 1 site we computed correlations between these parameters both for records of TT01 and CR1cores

(not shown in Figure 6). Any significant correlation between standardized $\delta^{18}$O and the number of HSD and the annual

hourly frequency of SWE was verified. The annual $\delta^{18}$O$_{TT01}$ presented a positive weak correlation with interannual HSD

variability (r = 0.28; p > 0.05 (0.29); α = 0.05) and negative weak with annual hourly frequency of SWE (r = -0.32; p > 0.05

(0.23); α = 0.05). In contrast, $\delta^{18}$O$_{CR1}$ presented negative weak relation with HSD amount variability (r = -0.22; p > 0.5

(0.49); α = 0.05) and no relation with frequency of SWE.


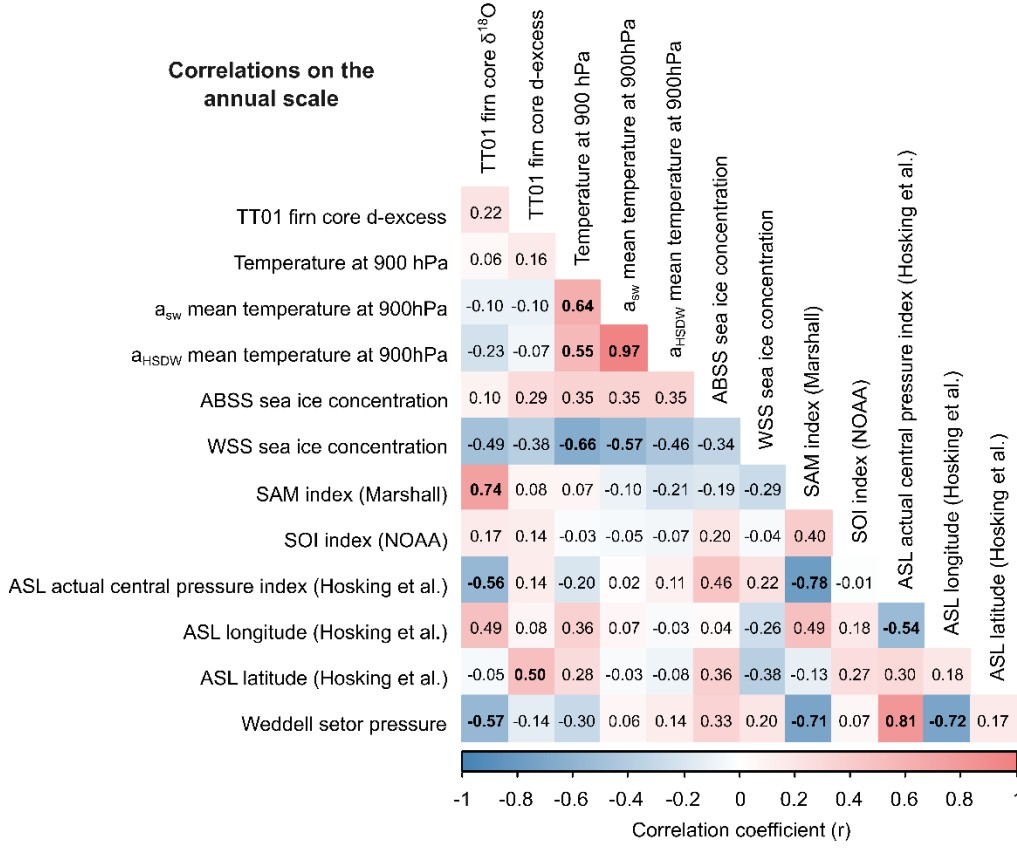

**Figure 6.** Linear correlation coefficients (r) table showing the relationships between isotopic time series of TT01 firn core and the climatic

parameters. Correlations on the annual scale (from 1999 to 2014) among the standardized average of $\delta^{18}$O and d-excess, the temperature at

900hPa, snowfall weighted mean temperature at 900hPa, HDS weighted mean temperature at 900hPa, ABSS$_{SIC}$ and WSS$_{SIC}$ average, SAM

[Marshall, 2003], SOI [NOAA] and ASL indices [Hosking et al., 2016], and mean sea level pressure in the WSS are presented in the





matrix. Statistically significant correlations (p < 0.05; α = 0.05) are marked in bold. The climatic time series were detrended, excluding the SAM and SOI time series.

### 3.5 Annual snow accumulation


      The mean annual snow accumulation rates derived from the TT01 core are similar for both the 1999-2014 (0.2429 ± 0.0037 m w. eq. yr$^{-1}$) and 2000-2011 (0.2441 ± 0.0058 m w. eq. yr$^{-1}$) periods. The average snow accumulation was 0.24 ± 0.07 m w. eq. and the TT01$_{snow\ accumulation}$ time series exhibited a non-significant slight decreasing trend (slope of -0.04; r = - 0.25; p > 0.05 (0.35); α = 0.05). Contrastingly, the mean snow accumulation rate derived from the CR1 core was 0.2781 ±

0.0112 m w. eq. yr$^{-1}$ for the 2012-2000 period. The average snow accumulation is 0.32 ± 0.13 m w. eq. and the CR1$_{snow\ accumulation}$ time series exhibited a non-significant slight increasing trend (slope of 0.04; r = 0.10; p> 0.05 (0.76)). Table 5 lists the snow accumulation rates (m w. eq. y$^{-1}$) for different periods computed of both cores (TT01 and CR1). Noticeably, the snow accumulation rates of the TT01 core had opposite behaviour in relation to the rates of the CR1 core, considering the three-time intervals analysed: 2000-2003, 2004-2008, and 2009-2012.


**Table 5.** Average snow accumulation rate for three different periods: 2009-2012, 2004-2008, and 2000-2003.

| Cores | Snow accumulation rate (m w. eq. y$^{-1}$) | | |
|:---:|:---:|:---:|:---:|
| | 2012-2009 | 2008-2004 | 2003-2000 |
| TT01 | 0.2179 (± 0.0187) | 0.2831 (± 0.0252) | 0.2417 (± 0.0247) |
| CR1 | 0.4112 (± 0.0231) | 0.1974 (± 0.0133) | 0.3771 (± 0.0445) |

      We obtained a local average annual snow accumulation of 0.24 ± 0.09 m eq. w. per year over 1999-2018 stacking

the AWS and firn cores data. For this, the snow height measurement obtained by the ultrasonic sensor installed near the Criosfera 1 AWS was multiplied by the average surface density (*i.e.*, by 0.37 g cm$^{-3}$; section 3.2.) to convert it to metres of w. eq. Composite record shows a significant slight decrease in accumulation from 1999 to 2018 (slope of -0.09; r = - 0.55; p = 0.01 and α = 0.05) (Figure S7 in Supplementary Material). Figure 7 presents a comparison among both annual average snow accumulation of the composite and individual records with the number of HSD per year, annual relative hourly

frequency of SWEs, annual average precipitation, and 10-m wind data from the ERA5. In the 1998-2018 period, there were on average 7.3 EPEs per year and 12.4 HSDs per year, and the EPEs tended to be more frequent in the 2013-2018 and 2002-2006 periods than in the 2007-2011 and 1999-2001 periods. We observed an anticorrelation (r = -0.52; p < 0.05 and α = 0.05) between the number of HSD and SWE over the 1999-2018 time span (Figures 7 and 8). Remarkably, the trend of the composite record was inconsistent with the trend of ERA 5 precipitation, but in some periods the snow accumulation tended





to approach the snowfall amount, such as 2015-2018 and 2002-2005. Furthermore, it was also consistent with the annual SWE frequency trend and inconsistent with the HSD frequency trend.

The observed annual snow accumulation over 1999-2018 related positively and moderately with annual SOI index ($r = 0.60$; $p < 0.05$ and $\alpha = 0.05$), concentration of sea ice in the WSS ($r = 0.51$; $p < 0.05$ and $\alpha = 0.05$), and weakly with mean pressure of the ASL and WSS (both with $r = 0.35$; $p>0.05$) (Figure 8). Further, it correlated negatively and moderately

with snowfall non-weighted mean temperature ($r = -0.42$; $p > 0.05$) (Figure 8). Significant, positive, and moderate correlations with summer ($r = 0.58$; $p < 0.05$ and $\alpha = 0.05$), autumn ($r = 0.44$; $p = 0.05$ and $\alpha = 0.05$) and spring SOI indices ($r = 0.50$; $p < 0.05$ and $\alpha = 0.05$) and no relation with winter SOI index ($r = 0.16$; $p>0.05$ (0.49) and $\alpha = 0.05$) were also verified (not shown in Figure 8). Running correlation made with a five-year window showed that the positive relationship between mean annual snow accumulation and annual SOI index was stronger in 1999-2003 and 2008-2011 ($r$ ranged from

0.59 to 0.95). Practically in the same period, the annual hourly frequency of SWE related positively and weakly to moderately with the annual SOI index (1999-2002 and 2007-2011; $r$ ranged from 0.12 to 0.58; except in 2001) and negatively and strongly to moderately with the SAM index (1999-2001 and 2006-2010; correlation coefficient ($r$) ranged from -0.47 to -0.89). Throughout the time span studied, the SWE hourly frequency presented a negative but not significant correlation with the SAM index ($r = -0.35$; $p > 0.05$ and $\alpha = 0.05$; Figure 8). Over 1999-2018, the increase of SWE frequency

is partially linked with the migration of ASL northward ($r = 0.68$; $p < 0.05$; $\alpha = 0.05$) and the relative increase of the pressure in WSS ($r = 0.62$; $p < 0.05$; $\alpha = 0.05$) and of the ASL ($r = 0.69$; $p < 0.05$; $\alpha = 0.05$; Figure 8). The number of HSD was positively and strongly linked with snowfall amount ($r = 0.73$; $p < 0.05$; $\alpha = 0.05$), and both were moderately related to temperature (Figure 8). Further, the variability of the HSD frequency and snowfall amount are also related negatively and moderately with ASL latitude ($r = -0.39$ for the snowfall-$ASL_{latitude}$ relationship and $r = -0.40$ for the $HSD_{number}$-$ASL_{latitude}$

relationship), although these relations are not significant ($p > 0.05$). At the Criosfera 1 site, the HSD contributed on average 42% to the annual precipitation (Figure S8) and it was more frequent in spring and less frequent in summer than in other seasons over the 1998-2018 period (Figure S9).

The comparison between Criosfera 1 AWS and ERA5 reanalysis data for 2013-2018 (except 2014; Figure 9 and S10-S12) revealed five key findings. First, the trigger of preserved annual snow accumulation at the Criosfera 1 site

presented some link with the distribution and intensity of the EPE throughout the year. As shown in Figure 9a and Figure S10, the EPE occurred more concentrated in the spring months and late winter and the preserved accumulation started in these periods in 2013 (Figure 9) and 2015 years (Figure S10). In contrast, in some years the EPE occurred in a more distributed way, both earlier and later in the year, and the preserved accumulation started in late summer or in autumn, such as in 2017 and 2018 (Figures 9 and S12). Specifically, in 2016, the start of the preserved snow accumulation was triggered

by an intense EPE event that took place in late summer (~24 mm w. eq. accumulated in four consecutive days of the high snowfall rate). Further, in the 2016 year, no additional accumulation occurred near the end of the year since HSDs happened majority concentrated in the middle of the year (Figure S11). Second, in years with a greater number of EPEs (*e.g.,* 2018; Figure 7), the cumulative snow accumulation pattern resembles the cumulative snowfall pattern (Figure 9).





Third, the snow deposited in Criosfera 1 site obviously precipitated a few metres to tens of kilometres south.
Adopting a wind speed threshold of 10 m.s⁻¹ for blowing snow and 5 m.s⁻¹ for snowdrift (based on Birnbaum et al., 2010 and Scarchilli et al., 2010 and references), undeniably there is an extra contribution of the long to short distance transported snow to accumulation and the snow erosion is a very active process given to the wind regime at Criosfera 1 site. Considering the ERA5 data, over 1999-2018 there was prevailing southerly wind throughout all years with an annual mean velocity at 850hPa and on the surface of 9.6 m.s$^{-1}$ and 10 m.s$^{-1}$ (Figure 7), respectively, which favour such processes (*i.e.*, blowing snow
and snowdrift) and their consequences (*i.e.,* additional deposition and erosion). For instance, the event in 2016 late summer yielded ~0.06 m of snow height but the snow height ultrasonic sensor measured an input of ~0.32 m and then a decrease of the accumulated snow in the following weeks until ~0.13 m (recorded before the next EPE) (Figure S11). Fourth, the accumulation pattern in Criosfera 1 site shows that non-EPEs are hardly preserved or probably are mixed with EPE records before settling down to Criosfera 1 site (Figure 9 and S10-S12).

Fifth, events that drive high snowfall rates at Criosfera 1 site come dominantly from the SE sector (Figures 9, S10-S12, and S13). They weaken the south wind and rarely exceed the 15.51 m/s at 850 hPa. *E.g.,* the wind velocity overcame this threshold during the high snowfall only in 2011 and 2012. Data of mean annual meridional wind anomaly show that air-masses incursions by WSS, from the Peninsula region to the east coast of the WSS, have intensified in the first two decades of the 21$^{st}$ century compared to the last century (Figure S14), and indicate that moisture comes primarily from this sector.
Further, westerly winds have been more intense in the 21$^{st}$ century than last century (Figure S14).





**Figure 7.** Comparison of annual average snow accumulation composite record with the number of HSD per year, annual relative hourly frequency of SWE (including only events with ws ≥ 15.51 m/s), annual average precipitation and 10-m wind data from the ERA5. The snow accumulation composite record was derived by averaging or linking the time series of Criosfera 1 AWS and the two cores (TT01 and CR1). The number of HSD per year and the annual relative hourly frequency of SWE were computed using ERA5 datasets. Individual records are shown at the bottom. The undulating curves are the best-fit polynomial trend lines to the sixth order.

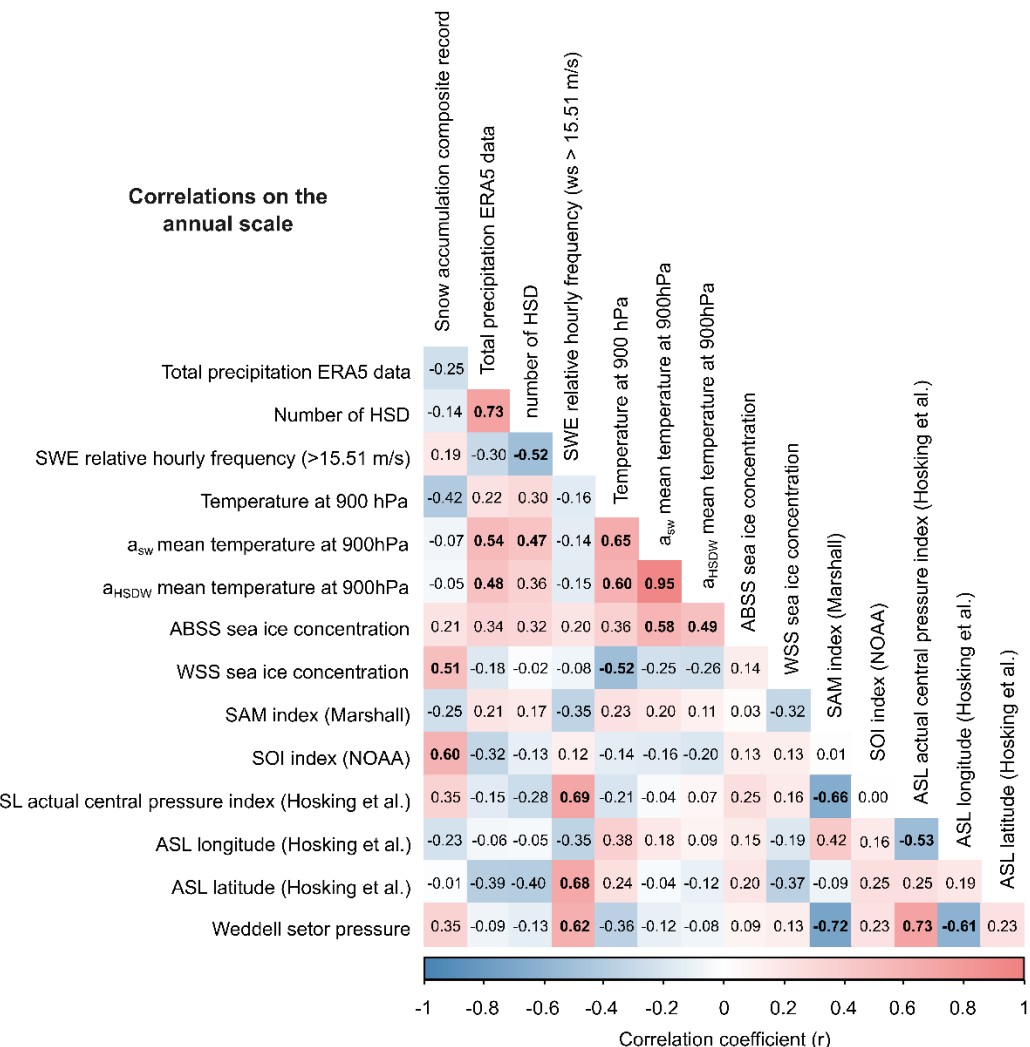

**Figure 8.** Linear correlation coefficients (r) between the composite record of snow accumulation from the Criosfera 1 site and climatic parameters: total precipitation ERA5 data, number of HSD, SWE relative hourly frequency, the temperature at 900hPa, snowfall weighted mean temperature at 900hPa, HDS weighted mean temperature at 900hPa, ABSS$_{SIC}$ and WSS$_{SIC}$ average, SAM [Marshall, 2003], SOI [NOAA] and ASL indices [Hosking et al., 2016], and mean sea level pressure in the WSS. Correlations on the annual scale (from 1999 to 2018) are presented in the matrix and statistically significant correlations (p < 0.05; α = 0.05) are marked in bold. The climatic time series were detrended, excluding the SAM and SOI time series.





(a)

**Figure 9.** Depositional history for (a) 2013 and (b) 2018. On the left: comparison among daily snowfall data from ERA 5 with records of daily snow accumulation and wind speed from the Criosfera 1 AWS. HSD is highlighted (blue-green dotted vertical line) and EPEs are indicated on the top left graphs. Cumulative daily snowfall and snow accumulation are shown on the bottom left graph. On the right: Relative hourly frequency of 850 hPa wind direction and speed derived from ERA5 datasets for both HSD (top right graph) and all days (bottom right graph). The adopted threshold for blowing snow and snowdrift is shown in the wind speed graph (thin grey line).



# 4 Discussion

We investigated the depositional and post-depositional history and the climatic significance of the stable water
isotopes and snow accumulation records preserved at the Criosfera 1 site. Therefore, we have explored the properties of
snow-firn pack stratigraphy, accumulation pattern, and a range of relationships between these derived chemical and physical
records from the firn cores and climatic parameters and indices. The results of our study are discussed below in an integrated
way.

Our results showed that the annual stable isotope ratios preserved at the Criosfera 1 site mark the annual SAM
variability (Figure 6), as well as in winter and spring. The SAM is the dominant climate variability mode from the Southern
Hemisphere, representing the circulation variability between high and medium austral latitudes as a function of the
atmospheric pressure gradient (e.g., Thompson & Solomon, 2002; Marshall, 2003; Gillet et al., 2006; Fogt & Marshall,
2020). Since the 1960s, the SAM has been in a positive trend, which became more intense from the 1980s onward, indicating
the strengthening and contraction of the polar jet and, consequence, the migration of the low-pressure zone southward
(Thompson & Solomon, 2002; Marshall, 2003; Gillet et al., 2006; Fogt & Marshall, 2020). Many studies have attributed the
later 20[th] century warming in Antarctica Peninsula to the positive phase of the SAM (SAM+), as well as the observed
cooling in EAIS in the same period (e.g., Kwok & Comiso, 2002; Thompson & Solomon, 2002; Marshall et al., 2006).
According to Noone & Simmonds (2002), the SAM (+) trigger an increased upward moisture flux in coastal and oceanic
areas around Antarctica due to a decrease of atmospheric pressure in these areas, which can isotopically enrich the
continental precipitation. Such a relation between the SAM and the additional moisture recharge near the coast explains the
significant positive correlations between the isotopic ratios and SAM at the Criosfera 1 site. Because SAM strongly
influences the temperature on the northeast side of the Antarctic Peninsula in all seasons (Clem et al., 2016), the isotopic
ratios from the Criosfera 1 site are also related positively to temperature in this region (Figure S6).

It is known that during the SAM (+) phase, pressures around the Antarctic continent generally decline. Also, the
SAM strongly influence the deepening of ASL, the main low-pressure climatological zone of Antarctica that is located in the
Pacific sector, causing an amplification of the advection of warm air masses into WAIS and Antarctica Peninsula and
providing a significant reduction in SIC in the Amundsen and Bellingshausen Seas (Turner et al., 2013; Hosking et al., 2013;
Raphael et al., 2016; Clem et al., 2017). This fact also supports the negative relationships observed between the isotopic
ratios of the Criosfera 1 site with the ASL central pressure and the WSS pressure (Figure 6). However, previous studies have
explained that cyclonic anomalies observed from 21[st] century in WSS are linked to shift to the negative phase of Interdecadal
Pacific Oscillation (IPO (-); which indicates persistency of La Niña-like sea surface temperature anomaly (colder) over the
tropical Pacific) (Turner et al., 2016). This new atmospheric circulation configuration (SAM (+) and IPO (-)) is identified as
causing the cooling observed in the Antarctic Peninsula region (Turner et al., 2016) and South Pole warming (Clem et al.,





2020) because of it intensifies the incursions of the air masses by WSS eastern coast and the expulsion of relatively colder,
denser continental air by Antarctica Peninsula.

Negative δs-WSS$_{SIC}$ and positive δs-near surface regional temperature correlations were observed (Figures 6 and
S6). Kumar et al. (2021) showed that in the phase of the SAM (+) occurred negative SIC anomalies in the western edge and
positive heat flux anomalies in the central and internal part of the WSS in both winter and spring. Nonetheless, it is known
that at SAM (+) level, a higher SIC tends to occur in all longitudes because of the anomalous divergence towards the north
driven by Ekman drift (Hall & Visbeck, 2002). Although such a SIC anomaly may not be directly associated with SAM,
Kumar et al. (2021) also observed a SIC negative trend in the eastern WSS region in winter (significant) and spring (non-
significant) from 1979 to 2019. Such features observed in WSS in recent years indicate an intensification of moisture input
to the atmosphere and increased atmospheric capacity to hold moisture in these seasons. Therefore, we conclude that the
increase in isotopic ratios may also be partially related to variations in SIC and temperature in the WSS and Peninsula
region. This conclusion is not only based on the pieces of evidence previously mentioned but also on the EPE bias on the
isotopic ratios at the Criosfera 1 site. For as observed, the EPES that reach the Criosfera 1 site come from the WSS (Figures
9 and S14), and they potentially occur from the middle to the end of the year (mainly in spring) (Figure S9).

The findings of Hoffmann et al. (2020) corroborate this interpretation regarding SIC influence. They showed that in
the Union Glacier area (UG, Ellsworth Mountain region, WSS), the isotopic ratios also had negative correlations with
WSS$_{SIC}$. However, there are some differences between our study and theirs. Hoffmann et al. (2020) found no correlation with
regional temperature in the WSS, instead they found with the temperature in the Dronning Maud Land region (hereafter
DML; Atlantic sector of EAIS). Further, they showed that the isotopic content preserved in UG from 1980 to 2014 was not
related to SAM. Two arguments can explain this observation: either (1) the UG is a climate transition region or (2) the signal
preserved in the UG is more strongly affected by the southerly wind than Criosfera 1 site. We observed that excessive noise
generated by wind influence could erase these relationships by evaluating the CR1 isotopic signal (Table S2). Indeed, the
katabatic winds strengthen in the UG given the topographical characteristics of the region and the nearness of the coast —
that is, the surface wind is intensified because of the influence of low-pressure centres around the continent over surface air
flux (Parish & Bromwich, 2007). Although these factors favour argument (2), some studies have shown that a transition zone
exists between ABSS and WSS near the UG (Nicolas & Bromwich 2011). In addition, also it is known that various large-
scale climate modes can influence the atmospheric circulation and, consequently, the precipitation of a given region in the
Antarctic continent (Marshall et al., 2017), which could explain differences in the relationship between isotopic content and
these patterns within the same sector. E.g., Marshall et al. (2017) depicted that the Pacific South American 2 (PSA2) pattern
justifies the climatic asymmetry between the Atlantic and Pacific sectors of the WAIS and places the UG area as a transition
zone. This pattern in its positive phase (PSA2+) corresponds to positive and negative pressure anomalies centred at 150°W
and 90°W, respectively (Mo & Higgins, 1998; Marshall et al., 2017). As per Ding et al. (2011), the PSA exerts interference
on the SAM structure in the Pacific sector, being responsible for the austral annular mode asymmetry, which may explain
variations in isotopic-SAM relationships throughout the WAIS. The fact that our results are not in line with those of



Roosevelt Island, Ross Sea Sector (i.e., negative correlations with SAM and positive correlations with ASL of similar magnitude; Emanuelsson et al., 2022), reaffirms this interpretation. Although the meridional wind anomalies observed in the 740 21st century resemble the PSA2 (+) pattern (Figure S14), only future studies correlating isotopic compositions with the PSA modes can further elucidate such relationships.

The composite record depicted a stable water composition increase in the 1999-2012 period at the Criosfera site, where only the increased d-excess was significant. Given the strong positive correlation of isotopic ratios with SAM, the increase in δs can be explained by the tendency of SAM to remain in its positive phase, as previously mentioned. While the 745 increase in d may be related to the moisture input coming from the sea ice zone. Bonne et al. (2019) mapped the vapour isotopic composition in the Atlantic sector, from Greenland to Antarctica. They showed that the isotopic composition of surface moisture tends to have a higher d within the sea ice zone. The explanation of higher d would be related to the sublimation of snow deposited on sea ice. However, the significant d increase in this period may be due to condensation of vapour with high d on the surface (Ritter et al., 2016) brought by the katabatic winds that come down the Antarctic Plateau 750 towards the coast of the WSS sector (Parish & Bromwich, 2007). As pointed out by some studies, isotopic exchanges take place between precipitation events in areas most affected by the south wind, such as in the Kohnen Station region (DML; Ritter et al., 2016) and in the Dumont d'Urville Coast (Pacific sector of the EAIS; Bréant et al., 2019). A piece of evidence that corroborates the last mentioned hypothesis is the positive and moderate correlation of ASL with the d-excess in the Criosfera 1 site since the increase in the frequency of SWEs is also linked to ASL migration to the north. Correlations with 755 sea ice in the ABSS and WSS hardly explain an increase in d-excess at the Criosfera 1 site. In addition, in periods with higher frequency SWEs, we noticed a positive trend of d-excess/δs slopes and d-excess-δs correlations in both cores not explained by features observed in the snow-firn pack, indicating an increase in d yielded by another process. Although SWEs are the potential explanation for the significant d increase between 1999 and 2012, further studies of vapour measurements in the sea ice area and at the Criosfera 1 site are still needed to explore further both hypotheses.

The annual accumulation preserved at the Criosfera 1 site between 1999 and 2018 had a significant and positive correlation with the SOI index, both annually and in the summer, spring and autumn. Furthermore, it also had significant and positive correlations with the pressure in the WSS, ASL central pressure, and the WSS$_{SIC}$. Such observations indicate that snow accumulation at the Criosfera 1 site tends to be greater during Lã Niña events (especially those that occur in the spring and summer) when pressures increase around the continent and the SIC increases in the WSS region. Kaspari et al. (2004) 765 compared the accumulation variability from various locations in the Pacific sector of WAIS and at the South Pole with the SOI index. From all the places they analysed, only the South Pole showed positive correlations with the SOI index, albeit weak. Furthermore, they proposed the existence of a transition zone of ENSO's influence somewhere between the South Pole and the Pacific Sector. Yet, Hoffmann et al. (2020) showed that the accumulation in UG had no relationship with ENSO, reinforcing the notion of asymmetry between the two sectors. The observed accumulation of 0.24 ± 0.09 m eq. w. per year in 770 the 1999-2018 period at the Criosfera 1 site confirms that the Criosfera 1 site is a region of moderate accumulation. However, a decreasing trend in snow accumulation was observed in the studied area during this period. Remarkably, this





may be due to an increase in the number of EPEs concurrent with a slight decrease in the incidence of SWEs. Over the 1999-2018 period, there was a significant and moderate anti-correlation between the frequency of HSDs and SWEs (i.e., there was an SWE-EPE seesaw relation). Moreover, it was found that the years with the highest accumulation at the Criosfera 1 site
(e.g., 2014-2006 and 2001-1999) were those where the frequency of SWE increased relatively. On the other hand, those with the lowest accumulation correspond to those with the highest number of HSDs (e.g., in 2015-18 and 2002-05). *I.e.*, in the period from 1999 to 2018, the increase in accumulation is linked to an increase in the speed of winds coming from the south and not necessarily an increase in snowfall. Yu et al. (2019) spatially correlated SWEs with SAM and ENSO from 1979 to 2017. According to Yu et al. (2019), SWEs negatively correlated with SAM in all seasons at the Criosfera 1 site, especially
in autumn, winter and spring, where these relations were significant. However, they found little relationship with ENSO. Our results confirm the negative relationship between SAM and SWEs (Figure 8) and indicate that it was stronger during periods where accumulation was highest. Furthermore, we showed that in the period 1999-2018, the increase in SWE frequency is related to the northward migration of the ASL (as previously mentioned) and to pressure increases around the continent, both factors that may be reflecting conditions of SAM (-) or weakening of SAM (+) phase. Although Yu et al. (2019) did not find
significant relationships with ENSO, our study shows that SWE-SOI relationships were positive although not strong in periods when higher accumulation occurred, which may justify the meaningful and positive relationship between accumulation and the SOI index.

The relative accumulation decrease observed in 20 years concomitant to the number of HSDs increase (Figure 7) can be explained by the fact that the occurrence of EPEs tends to decrease the speed of the south wind (the factor that plays a
role principal influence on the accumulation in the studied period, as previously explained) in the Criosfera 1 site. The proximity of snowfall can be attributed to the increased difficulty of the wind in eroding large amounts of snow. Observations of snow height and wind speed recorded in Criosfera 1 AWS and snowfall reanalysis data between 2013 and 2018 support such an interpretation. Our analysis focuses on the transition from a scenario where EPEs were less frequent (2013) to a scenario where EPEs were more frequent (2018). During this period, we noticed that the greater the number of
EPEs, the greater the probability of the accumulation pattern approaching the snowfall pattern (case of the 2018 year). In addition, we realized that EPE events tended to be preserved while non-EPE to be swept away. We extend the interpretation that mainly EPE signal is preserved until the 1999 due to seasonal bias of the isotopic records from the Criosfera 1 site. An explanation for accumulation 2015-2018 has approached even closer to snowfall than in the 2002-2005 period is that in the former period, there was an intensification of negative sea ice anomalies in the WSS linked to depressions increase in WSS
(Turner et al. 2020), which could have interfered in the intermittence of the EPEs. Our analysis of the seasonality of HSDs may support such an interpretation (Figure S9).

Turner et al., 2019 showed that the number of EPEs has increased slightly in recent decades in the WSS (considering the period 1979-2016). They also illustrate that the frequency of EPES was positively correlated with SAM in the WSS, and this relation was significant in many areas of this sector, especially in spring, summer and autumn. This
evidence, together with the fact of HSD frequency and snowfall amount have presented some link to migration of ASL





southward justify the establishment of an SWE-EPE seesaw relation in the 1999-2018 period. Although SAM does not explain the local HSDs number (Figure 8), the frequency of these extreme events has increased slightly in recent years at Criosfera 1 site (Figure 7). This increase is consistent with the positive trend in SAM in recent years and with the observations of Turner et al. (2019). In addition, this increase is also consonant with the ASL's tendency to move southward in recent years (Hosking et al., 2016). These observations support the interpretation that in a future scenario of atmospheric warming and persistency of SAM (+), the highest accumulations will be related to EPEs and no longer to SWEs.

Our study also highlighted that climatic signatures also can be extracted from the stratigraphic features and properties. Framework changes in the snow-firn pack from Antarctica inland are mainly driven by local wind and temperature regimes besides the thermodynamic instability and vertical strain (Sturm, 2003; Barry & Gan, 2011). For instance, both these climatic features induce temperature gradients in the surface snowpack. When the temperature gradient is slight-moderate, it is established no strong vapour gradients and occur continuous rounding, densification, and sintering of ice crystals. Nonetheless, when a strong temperature gradient establishes in the snow-firn pack, it yields more intense vapour transport favouring kinetic metamorphism. A typical feature resulting from the kinetic growth metamorphism is the depth hoar (Singh et al., 2011).

The analysis stratigraphic in Criosfera 1 site shows that depth hoars are potential features to mark intermediate seasons for coinciding with the transition between maximum and minimum delta peaks (Figure 3). It also reaffirmed that in Criosfera 1 site, temperature contrast was common in the studied period. Pinto (2017) already had shown that spring is the season in which surface temperature varies widely in a short time at Criosfera 1 site. These observed conditions put the spring as the main season for depth hoar formation. However, our data suggest that autumn is another important season and does not rule out the possibility of depth hoar growth in winter (Figure 3). Since depth hoar is primarily formed in the thinner layers (Singh et al., 2011), differences regarding the density of depth hoar between TT01 and CR1 core in correlate periods can be explained by the latter presenting a higher accumulation rate than the former.

In previous studies, ice lenses were considered to occur typically in summer when the temperature reaches the ice melting point essentially in coastal areas around the continent and Antarctica Peninsula (King & Turner, 2009), being indicated like summer markers (Cuffey & Paterson, 2010). Nonetheless, surface melting in winter and some melting in summer have been recently associated with atmospheric river events in both inland and coastal areas (Wille et al., 2019). The surface melting yield in winter is justified by the presence of mixed clouds that may provide downward longwave radiation able to generate melting, as well as by the föhn wind on the leeward side of the mountain range due to a combination of sensible heat and downward longwave radiative fluxes (Wille et al., 2019). At the Criosfera 1 site, it was observed ice lenses in both winters (match with low δ) and summers (match with high δ), which raises suspicions about atmospheric river influence. However, more stratigraphic studies are necessary to verify if some of these observed thin melting features at the Criosfera 1 site are related to atmospheric river events.

Although our study shows clearly that both isotopic and accumulation records from the Criosfera 1 site imprint both large-scale (SAM, ENSO, and possibly PSA2) and synoptic (EPEs and SWEs) influences, we also observed some effects





that potentially can obliterate the climate signal. Differences in accumulation between both cores imply that topography is
undulated. Since accumulation spatial variations are strongly linked to changes in surface slopes, with more deposition
occurring in lower areas than higher (Frezzotti et al., 2004; Kaspari et al., 2004; Frezzotti et al., 2013), probably CR1 and
TT01 lied near a through and a ridge, respectively. Even though no topographic study has been yet performed at Criosfera 1
site, surely the topographic effect is the cause of the noisier signal verified on the CR1 core, which led to both losses of the

correlation between δs and climatic features and contradictory δs and EPEs relation. Furthermore, we evaluated that due to
SWE-EPE seesaw relation and intermittency effect (EPE biased snowfall) the reconstructions of the local temperature are
unfeasible and, therefore, attempts to build a local thermometer, as that executed by Bezerra (2016) are not indicated. To
carry out WSS climatic reconstructions on the secular scale using records from the Criosfera 1 region, we recommend the
recuperation of new cores and spatial sampling in pits to construct composite records more robust and minimize or eliminate

the non-climatic bias of the isotopic and accumulation signal.

## 5 Conclusions

In this study, we evaluated the stable water isotope compositions and accumulation records from the upper reaches

of the MIS basin (Criosfera 1 site; ~84°S, ~79° 30'W; WSS) in order to assess the depositional history and examine which
climatic information is stored on this site. We found that over the 1999-2014 period, the interannual δs variability is strongly
explained by variations of SAM level and by SIC anomalies in the Weddell Sea to a lesser extent. Although δs records from
the Criosfera 1 site do not capture local temperature variations due to local post depositional interference associated with the
strong wind regime in this site, they respond to regional temperature changes in WSS and Antarctic Peninsula. This evidence

indicates that the records of the upper reaches of WSS are suitable for temperature reconstructions of these regions. We
conclude that these good relations observed between δs and climatic features and SAM at Criosfera 1 site are because the
signal is biased by EPEs coming from WSS. EPEs are the trigger to start accumulating the preserved snow, but an increase
in its frequency is not responsible for higher annual snow accumulation in the studied area over the 1999-2018 period.
Interesting, our snow accumulation composite record shows that the SWE-EPE seesaw governs the snow accumulation in

the upstream area of the MIS basin in the studied period: where the higher accumulation tended to occur in a period of
(slight) increased SWEs and the lower accumulation in a period of (slight) increased EPEs. It seems that the observed SWE-
EPE seesaw at Criosfera 1 site is partially explained by SAM level variability but also has some tropical influence since the
SOI index explains partially the interannual variability of the snow accumulation.

Our results suggest that both isotopic compositions and snow accumulation are strongly influenced by large-scale

modes of climate variability (SAM, ENSO and possibly PSA) and synoptic scale events (EPEs and SWEs) in the MIS basin
inland. Furthermore, they also provide valuable information to understand mass balance at the basin scale in the WSS and
stress that higher snow accumulation can reflect windier periods than a snowfall increase. The latter evidence has
implications for the interpretation of accumulation reconstructions in other ice core areas. The observations of our study point out that in a scenario of future warming, the persistence of SAM positive trend, and the EPE increase due to
intensification of wetter and warmer air masses incursions by the WSS will cause a change of the main driver of the snow accumulation. We recommend more shallow drills and snow pits in this site to build the best composite record to reconstruct these atmospheric circulation patterns and solve challenges regarding the topographic effect.

*Competing Interests*

The authors declare that they have no conflict of interest.

*Acknowledgements*

Andressa M. de Oliveira thanks the Coordenação de Aperfeiçoamento de Pessoal de Nível Superior (CAPES) for the master's scholarship. This work was accomplished with resources from the Conselho Nacional de Desenvolvimento
Científico e Tecnológico (CNPq, Process no. 465680/2014-3 - INCT da Criosfera). We thank the support of our colleagues Filipe G. L. Lindau and Luciano Marquetto during the fieldwork.

*Author contributions*

Conceptualization, A.M..; ice core drilling, R.T.B., F.E.A., J.C.S.; ice core decontamination and melting, R.T.B., I.U.T., 
F.E.A.; stable isotope analysis, A.M., R.T.B.; trace ion analysis, A.M., I.U.T.; additional methodology, A.M.; software, A.M.; validation, A.M.; formal analysis, A.M.; investigation, A.M.; data curation, A.M., R.T.B., F.E.A., P.T.V., V.S.; writing—original draft preparation, A.M.; writing—review and editing, J.C.S., R.T.B., P.T.V.; supervision, J.S.; project administration, J.S.; funding acquisition, J.S. All authors have read and agreed to the published version of the manuscript.

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
