# Peer review of "The stable water isotopes and snow accumulation from Weddell Sea sector imprint the large-scale atmospheric circulation variability"

_The Cryosphere, 2022_

## Author Comment (AC1)

Dear Anonymous Referee #1,

We thank you for your time to read and expertise to evaluate our article. Furthermore, we thank you for all the helpful suggestions. The responses to your questions and comments are given below. Please find our responses in *italics*, while we kept your original comments in normal text. Note that some highlights are given in red.

We hope that misunderstandings have been solved. Further, we apologize if we misinterpreted some of your questions.

**General comments:**

The authors present new stable water isotope and accumulation data of two firn cores extracted from the Möller Ice Stream in the Weddell Sea Sector (WSS) of Antarctica. They investigate the temporal variability of the isotopic and accumulation records and their relationship to meteorological station and reanalysis data and large-scale atmospheric modes. They identified temporal changes in the phase of the Southern Annular Mode (SAM) and associated pressure changes of the Amundsen Sea Low (ASL) and the WSS as well as temporal changes in sea ice concentrations in the WSS as the most important factors influencing interannual isotope variability. Furthermore, the authors attempt to reconstruct the depositional history of the study site. They suggest that strong wind events (SWE) and extreme precipitation events (EPE) are the main drivers of local snow accumulation, acting like a sea saw, i.e., accumulation increases when SWE frequency increases but EPE frequency decreases and vice versa. They emphasise the importance of their records for reconstructing atmospheric circulation patterns in the WSS, for understanding the mass balance of the Möller ice stream and how this might change under a future climate warming scenario.

There is no doubt about the uniqueness of the presented datasets and their potential to contribute to a better understanding of cryosphere-atmosphere interactions in the Weddell Sea Sector and Antarctica in general. The authors put a lot of effort in analysing the derived isotopic and accumulation data and understanding the reasons for observed temporal changes in the records. However, I have a little doubt that the manuscript is suitable for publication in its current state, unless major revisions are made.

There are two main reasons for this:

In general, it was quite exhausting for me to read through the manuscript, since it is very long, mainly because the authors lose themselves in the details. The methods are described very much in detail, but many of the provided information is not necessary to understand the records and the story of the paper. In contrast, the discussion section has no subsections, which makes it very difficult to follow the authors' thoughts. Better structuring of the discussion section is of major importance for the improvement of the manuscript. In addition, results and discussion

are mixed up in some parts of the manuscript. However, all these issues regarding the structure of the paper can be more or less easily fixed and are not an obstacle to the publication of the manuscript.

> ***Answer:*** *Dear reviewer, we eliminated some details previously described in our methodology section. Regarding the discussion section, our answer is given in the following (I.e., at the beginning of the issues about the discussions). However, if you find it strictly necessary to divide the discussions into subtopics for publication, we will do so in the order in which the results are presented to make it easier to read.*

What troubles me more are the correlation analyses of isotopes and snow accumulation, respectively, with meteorological records and climate indices. There is nothing wrong about finding no correlations or trends in your data, but non-existing trends and statistically insignificant correlations should not be stated explicitly, i.e., by listing all r- and p-values, in the text. To read through all the insignificant numbers makes it very hard to follow the story. Honestly, I got a bit confused by all the correlations, partly because there are many not statistically significant ones that are described too much in detail. It also makes no sense to discuss such insignificant relationships. If the p-value is much larger than your α-value, there is no correlation or trend that can be reasonably discussed, even if the correlation coefficients are something like ± 0.3 or ± 0.4. The "high" correlation coefficients may only have occurred by chance.

> ***Answer:*** *We carefully try to eliminate all non-significant trends and correlations from the text of our article as you can see in the answers below.*

**Specific comments:**

Title: Rephrase to "Stable water isotopes and snow accumulation from the Weddell Sea sector imprint the large-scale atmospheric circulation variability"

> ***Answer:*** *Sure. Title rephrased to…* ***"Stable water isotopes and snow accumulation from the Weddell Sea sector imprint the large-scale atmospheric circulation variability"*** *… according to your suggestion.*

Abstract: The abstract is too long and detailed. Even if there is no limit to the number of words, I think you should try to shorten it to a maximum of 250-300 words. Do not state numbers like coordinates, correlations with p-values in the abstract. Also, abbreviations like AWS and δs are not explained. If just reading the abstract, I would wonder why the isotopic and accumulation records are of different length, although they are derived from the same cores.

*Answer: In light of your comment, we have deleted 154 words from our summary. We also reformulated some sentences and explained the requested abbreviations (e.g., δs). The sentence that contained the AWS abbreviation was deleted to reduce the length of the abstract. Coordinates, correlations with p-values were also excluded. Therefore, our abstract changed to:*

*"Stable water isotopes and accumulation data extracted from polar ice/firn cores provide valuable climate information. Here, we present novel isotopic and accumulation time series from an upstream area of the Möller Ice Stream (MIS) basin, Weddell Sea Sector (WSS), Antarctica. Our purpose was to understand the depositional history and investigate how much the recent climate signal (21st century) is stored in the shallowest ice sheet layers in this area. Therefore, we investigated the relationship between (δs (i.e., δ18O and δD) and d-excess) and snow accumulation data of two shallow firn cores (both ~9.0 m deep) and glaciological information, local and regional meteorological data, indices of large-scale atmospheric modes (as SAM and ENSO) and the Amundsen Sea Low (ASL). We find that interannual δs variability is strongly explained by changes in the phase of the SAM and, consequently, also by changes in pressure of both the WSS and the ASL. Further, the regional temperature in WSS is another factor that measurably influence the δs. In the period covered by our study, the rarest and strongest wind events (SWE; > ~15 m/s) and extreme precipitation events (EPE) oscillate almost completely out of phase, and this oscillatory pattern justifies the non-temporally stable correlation between δs and local temperature in the studied area. For the period of 2013-2018, we show that the trigger to start accumulating snow on the studied site is the occurrence of a range of EPE in a short time or of the one EPE with higher snowfall rates and that, the low snowfall events are hardly ever preserved. Our snow accumulation composite record shows that the SWE-EPE seesaw governs the snow accumulation in the upstream area of the MIS basin in the 1999-2018 period. When the frequency of SWE increases and EPE decreases, the local snow accumulation increase. In contrast, in the opposite scenario, the accumulation approaches the forecast precipitation data indicating that the influence of blowing snow and wind drift decreases. Because of this relation, incredibly there was a significant decrease in snow accumulation in the study area in the 1999-2018 period due to an increase in EPE in recent years. Probably, in a scenario of future warming, the persistence of SAM positive trend, and the EPE increase due to intensification of wetter and warmer air masses incursions by the WSS such a relationship will change. Our results indicate that both isotopic compositions and snow accumulation are strongly influenced by large-scale modes of climate variability in the MIS basin inland. Furthermore, they also provide valuable information to understand mass balance at the basin scale in the WSS."*

**Major comments:**

**INTRODUCTION**

**P4 L120 ff.:** This paragraph already provides a detailed description of your methods. It is ok to give an overview of your methods in the introduction but keep it short. Do not explain each analysis step here as this belongs into the methods section.

*Answer: We thought that our description regarding methods in the Introduction was not so long. However, we understand your point and, therefore, we changed the last paragraph to…*

*"Challenges in interpreting both isotopic and accumulation records are amplified by the shortage of these records in various Antarctica sectors. Because of this and given the importance of these*

*proxies for climatic reconstructions, shallow core studies are encouraged to know which and how much climate information can potentially be extracted from isotopic and accumulation records on a basin-scale in each Antarctic sector (Masson-Delmotte et al., 2008; Thomas et al., 2017; Goursaud et al., 2019; Marcher et al., 2022). Here, we investigate the stable isotopic content (δ18O, δD, and d-excess) and the snow accumulation variability from two high-resolution shallow firn cores drilled on the upper reaches of the Möller Ice Stream basin, Weddell Sea Sector, in the transition between the West Antarctic Ice Sheet (WAIS) and the East Antarctic Ice Sheet (EAIS). We aim (1) to understand the depositional history (i.e., explore the annual accumulation pattern, probable events preserved, and post-depositional processes), and (2) to examine which and how much recent climate information (from the late 1990s onwards) is preserved on this site. For this purpose, we reconstructed the annual snow accumulation rates, investigated the precipitation intermittency, and evaluated how the proxy record is constructed. We explored the relationships between annually averaged snow accumulation and the number of HSD, frequency of SWE, the temperature at 900hPa, sea ice concentration (SIC) both from Amundsen Bellingshausen Sea sector (ABSS) and Weddell Sea sector (WSS), large-scale atmospheric modes (as Southern Annular Mode (SAM) and El Niño-Southern Oscillation (ENSO)), and pressure in both Amundsen Sea Low (ASL) and WSS. The same relationships were yielded for the annually averaged stable isotope composition time series to assess its climatic significance. We also considered the impact of the post-depositional processes driven by the wind and strong temperature gradients in the snowpack on isotopic compositions, as well as the influence of topography on both isotopic compositions and accumulation records. This study contributes to the understanding of the stable water isotope compositions and snow accumulation records from the WSS inland. It also points out the potential of these records to reconstruct the large-scale circulation variability."*

*… to minimize a possible "exaggerated detail" on methods in this section. The main modifications are highlighted in red colour (here two sentences were deleted, and one was reformulated).*

**MATERIALS AND METHODS**

Your methods section comprises almost 10 pages and you provide a lot of details that are not necessary to understand how you derived your records. In general, it is good to describe the methods carefully, but all the tiny details you give make this chapter very tiring to read. If you really want to provide all these details, consider putting some of them into the Supplementary.

*With the corrections and conceding some of your suggestions listed below, we succeeded in reducing a little without the use of Supplementary Material.*

P 5 L 146 It is not important where the name Criosfera comes from. Skip this information.

*Answer: Ok. We changed the sentence to…*

*"In the Criosfera 1 is installed a Brazilian AWS (at 84° 00' 00'' S, 79° 29' 39'' W), which is been in operation since January 2012, providing information about solar radiation, 2-m air temperature, wind speed and direction, relative humidity, snow height, among other environmental parameters (Pinto, 2017)."*

P 5 L 156: I am a bit in doubt that you can determine the ice thickness with such a high precision even if the BEDMAP 2 dataset might provide this number. Round it to 1900 m or at least to 1870 m.

Figure 1: Same as in the text. I suggest rounding the ice thickness.

*Answer: This data was extracted from the BEDMAP 2 dataset hosted in the Quantarctica project (QGIS). We are sorry, but we believe that changing is not necessary as it is the value given by the database and, probably this data has an associated error, then it is indifferent to round it. In addition, the data as it is presented does not alter the interpretations made in our study in any way. It is just information about the studied site.*

P 6 L 161: Which changes do you mean with "such changes"? Be more specific.

*Answer: We meant about changes in heat transport. To be more specific, we changed the sentence to…*

*"However, climatic models point out that in a scenario of future warming and changes in heat transport in the oceans, this basin along with the Institute Ice Stream basin would be the basins of WSS most affected by changes of meridional heat transport (Siegert et al., 2019)."*

P7 L 185: Where does this depth of 106.11 m come from? Table 1 says that CR1 is 9.13 m deep.

*Answer: We understand your question. Perhaps it wasn't so clear that the CR1 ice core has in fact 106.11 m. However, in our study, we only analyzed the first 10 meters.*

P 7 L 195ff. These are again a lot of unnecessary details. Shorten here (e.g., it does not really matter for the story of the paper whether samples were transported by air or road).

*Answer: Of course. To minimize the "story" we proposed to condense the paragraph as shown below…*

*From L200 to L206:*

*"The pieces of both cores were weighed on a lab scale (Ohaus®; precision: 0,01g), and then densities were determined using the core diameter, length, and weight measurements. In addition, an accurate description of the structures and properties of firn/ice was carried out for stratigraphic studies. All pieces were packed in a polyethylene bag and stored in the high-density Styrofoam box by protocol. From Antarctica, the cores were transported frozen to the Climate Change Institute (CCI), University of Maine (UMaine; USA), where it was decontaminated and subsampled in certified class 5 cold and clean rooms (ISO 14644-1, 1999)."*

P 8: I suggest combining 2.3.1 and 2.3.2 with chapter 2.2 into one chapter called "Firn/ice core collection and processing". The method descriptions you provide in chapters 2.3.1 and 2.3.2 are very detailed. Simply describe what was measured with which instrument. All the details on how the instruments work or which water was used for rinsing are unnecessary for the story of the paper. You do not write a paper on methodology. If you want to keep all the information in the paper, consider putting them into the Supplementary.

*Answer: We combined subsections 2.3.1. and 2.3.2. and reformulated the title. We have deleted some information about the CMDS instrument, transport story, and drilling details. The new section is given below…*

*2.2 Firn/ice core collection and processing*

*The CR1 ice core (106.11 m depth) was drilled in the 2011-2012-austral summer (between January 2nd and 10th) during the 29th Brazilian Antarctic Operation (Criosfera 1 AWS install campaign). A team of Brazilian researchers from the Centro Polar e Climático of the Universidade Federal do Rio Grande do Sul (CPC/UFRGS) and the National Institute for Space Research (INPE) were responsible for drilling. The CR1 core was obtained roughly 70 m east from the Criosfera 1 AWS (at 83° 59' 59.1'' S, 79° 29' 19.3'' W; Figure S1 in Supplementary Material) with a Fast Electromechanical Lightweight Ice Coring System (FELICS; Ginot et al. 2002), which performs one hole drilling without the need for trench excavation.*

*The TT01 firn core was recovered at ~30 m east from the Criosfera 1 AWS (83° 59' 59.5'' S, 79° 29' 31.4'' W; Figure S1 in Supplementary Material), in the Brazilian Traverse to WAIS (hereafter, BR-WAIS traverse), on January 7th, 2015. BR-WAIS traverse was carried out in the 2014-2015 austral summer (between January 4th and 21st) during the 32nd Brazilian Antarctic Operation and was attended by a research group from the CPC/UFRGS. See Marcher et al. (2022) for more details about this traverse. A Mark III auger (Kovacs Enterprises, Inc., USA) coupled with an electrical drill drive powered by a generator was used to collect the TT01 core. Unlike FELICS, it was necessary to construct a pit (2 m deep) to advance the drill to the deeper levels.*

*The pieces of both cores were weighed on a lab scale (Ohaus®; precision: 0,01g), and then densities were determined using the core diameter, length, and weight measurements. In addition, an accurate description of the structures and properties of firn/ice was carried out for stratigraphic studies. All pieces were packed in a polyethylene bag and stored in the high-density Styrofoam box by protocol. From Antarctica, the cores were transported frozen to the Climate Change Institute (CCI), University of Maine (UMaine; USA), where it was decontaminated and subsampled in certified class 5 cold and clean rooms (ISO 14644-1, 1999).*

*Both cores were decontaminated in a class 5 cold room at a temperature below -20°C using the method of Tao et al. (2001). A brief description of the methodology used is given below. Thin layers on the sides (4 mm thick) and ends (2 mm thick) of each piece of firn were removed using a pre-cleaned ceramic (ZrO) knife under a clean bench with a laminar flow system. Visual description of the stratigraphy was then enhanced. Pieces were conditioned in pre-cleaned acrylic tubes before being moved to another clean room for melting. All labware used was carefully pre-cleaned with acid baths and rinsed with type I water from the Milli-Q system (resistivity > 18 MΩ cm). During the decontamination step (and also during melting and subsampling; steps detailed below), Tyvek® protective suits (DuPont™, Wilmington, DE, USA) and polyethylene (PE) gloves were worn to reduce contamination.*

*The cores were melted using a continuous ice melting system with discrete and high-resolution sampling (CMDS) developed by CCI researchers (Osterberg et al., 2006). This system is installed in a class 5 clean room with high efficiency particulate air (HEPA) filters. See Osterberg et al., (2006) for more details about this melting method. In short, the samples for isotopic analysis were extracted from the outer zone of the core and stored in 25-ml vials of high-density polyethylene (HDPE), whereas samples for ion chemical analysis were extracted from the inner zone of the core and stored in 5-ml vials of polypropylene (PP). Before use, the PP vials were triple rinsed in type I water, soaked in type I water for 24 hours, triple rinsed again in type I water, and dried under a class 5 clean bench as recommended by Curran & Palmer (2001) and Osterberg and co-authors (2006). The CMDS system is always washed with ultrapure water before and after the melting step, and water blanks are collected.*

*Ice samples from both cores remained stored in a cold storage facility in Maine (USA) until 2018 when they were sent to CPC/UFRGS (Brazil). There, they were kept frozen until analysis.*

P 9: I suggest combining chapters 2.4.1 and 2.4.2 into one chapter 2.4 called "Snow isotope and ionic chemistry analysis".

**Answer:** *Ok. We combined these two subsections. A brief overview of the new section is given below…*

*The refrozen samples of the two cores were analysed at the laboratories of the CPC/UFRGS (Brazil). Table 2 summarizes the analytical method used for each core, the chemical parameters, and the number of samples analysed.*

*[…]*

*The stable water isotope analysis was performed at the Stable Isotope Lab using two wavelength-scanned cavity ring-down spectroscopy (WS-CRDS) systems (PICARRO® L2130-i, USA) — one coupled with a Combi PAL Autosampler (CTC Analytics AG, Switzerland) and the other coupled to a Picarro Autosampler (PICARRO® A0325, USA). …*

*[…]*

*All trace ions were expressed in concentrations of microgram per litre (µg L−1). For quality control, the composition of the ultrapure water yielded by the laboratory's Milli-Q system was routinely verified. Blanks from the melting step were also analysed, and their mean results were calculated for subtraction from the analytical datasets (according to Thoen et al., 2018). In this paper, we use only $Na^+$, $Cl^-$, and $SO_4^{2-}$ (also non-sea-salt (nss) $SO_4^{2-}$, which is introduced in the next section) datasets.*

P 10 L 267: Rephrase: "Precision of the measurements was… "

**Answer:** *Ok. We changed the sentence to…*

*"Precision of the measurements was better than 0.9‰ and 0.2‰ for $\delta D$ and $\delta^{18}O$, respectively."*

P10 L 274 ff.: You already list the measured ions in Table 2. Just refer to the Table here and do not list everything again in the text.

> *Answer: Sure. It was corrected.*
>
> *"[…] The trace ion species measured in this study are listed in Table 2. […]"*

I also think that the details provided on the instrument are not necessary to understand what you did.

P 10 L 281 ff. Same as before, too many details. Just state that you used the methodology of Thoen et al. with slight modifications and refer to the Supplementary, where you can provide all the details.

> *Answer: We sorry, but in this point, we disagree with your opinion. In our view, our level of detail is in line with other articles on the same subject. Furthermore, **we** think it is not necessary created a topic in our supplementary material to explain that the only difference between our method and Thoen's is that we modify the eluent generation gradients.*

P 10 L 293: You have already provided this information in Table 2.

> *Answer: Ok. However, in this case we think that table and figure captions are often not noticed, so we think it is worth reiterating the information here.*

P 11: Replace "estimates" by "estimation" in the title of chapter 2.5

> *Answer: Ok. We inserted estimation in the title as...*
>
> *2.4 Dating of firn cores and estimation of annual snow accumulation rates*

Please provide a reference for equation 2.

> *Answer: We added some references (in red) ...*
>
> *We assumed that Na+ was essentially derived from sea salt (ss) and estimated the nssSO42- using the well-known formula (Eq. 2; Abram et al., 2013):*
>
> *$[nssSO_4^{2-}] = [SO_4^{2-}] - 0.251 \times [ssNa^+]$ (Eq.2)*
>
> *where 0.251 is the [SO4−/Na+] weight ratio in seawater (Wilson, 1975) and SO42− and ssNa+ are the concentrations of these ions in snow/firn samples.*

P11 L 309: What do you mean with "others"? Either be precise or delete this.

      *Answer: "Others" was deleted.*

P11 L 315: Better use 1st January, than "New Year's Day".

      *Answer:  Ok. We exchanged "New Year's Day" for "1^{st} January"*

P11 L 320f.: Which cores did you use? Be precise and provide references if applicable.

      *Answer:  We inserted the references as shown below in red...*

      *In order to increase the accuracy of ACL dating, we used the $nssSO_4^{2-}/Na^+$ ratio along with the $nssSO_4^{2-}$ data and global volcanism historical records (available at: https://volcano.si.edu/; last access: May 2022) to identify explosive volcanic events and compared the ▧▧D and $nssSO_4^{2-}$ trend from TT01 with that from another firn core drilled 7.5 km west of CR1 AWS (Lindau et al., 2016). We also seek support in other cores from the WSS (as PASO-1 firn core (Hoffman et al., 2020) and BR-IC-4 firn core (Lindau et al., 2016)).*

P11 L 325f. Delete that the approach was challenging and provide the dating uncertainty for both cores instead.

      *Answer:  Ok. As suggested this information was deleted. Regarding uncertainties, these are found in the Results section (3.1.).*

P 12 L 342 What does "satisfactorily" mean? Can you quantify that?

      *Answer:  The answer to your question is in Table S1, as referenced in the sentence...*

      *"In addition, the ERA5 satisfactorily reproduces the variability of the meteorological parameters at the Criosfera 1 site, although overestimates the temperature and underestimates the wind velocity (see Table S1)."*

      *Please, if you could, take a moment to check the Table S1.*

P12 L 343f. How do you conclude from the correlations provided in Table S1 that ERA5 overestimates/underestimates the temperature/wind velocities? I do not understand how you can read this information from correlations without looking at the absolute values.

*Answer:* There was a misunderstanding, we thought. We never mentioned that it is the correlations that indicate if there is an overestimation or underestimation. In fact, it is the inclinations listed in the table that bring this information. We apologize as we thought it was implied.

P 14 L 396: What do you mean with "mean pressure at the level"? Which level?

*Answer:* We sorry by the missing word. We changed the sentence to...

"The same climatic indices and parameters were also correlated with firn core isotopic records. The series of precipitation, temperature, SIC, mean pressure at the sea level, number of HSD, and frequency of SWE used in the correlations were detrended."

**RESULTS**

P 14 L 406: Did you consider calculating diffusion lengths for your cores. Then you can better quantify the possible effects of diffusion on the isotopic values.

*Answer:* We are sorry, but deeply evaluating the diffusion in the Criosphere 1 area is not the focus of our work. Perhaps such a suggestion would be interesting for a future study using more cores drilled in this area.

P14 L 419: What is nssSx? Please rephrase the whole sentence. I guess you want to say that you detected peaks in both cores which are correlated.

*Answer:* We corrected the sentence as shown below:

"In 2006, it was detected peaks of $nssSO_4^{2-}/Na^+$ in the TT01 and of $nssS/Na^+$ in the PASO-1 firn core (drilled at 79°38'00.68''S, 85°00'22.51''W; near the Ellsworth Mountains; Hoffmann et al. 2020), which are correlated."

P15 L 425/L431: Please provide a unit for the dating uncertainty (I guess year?).

*Answer:* Correct. The uncertainty was given in years.

P 17 L 446: Use plural here: "… the minima and maxima of the cores…"

> *Answer: The plural was corrected.*

> *"However, the minima and maxima of the cores mismatch over time (Figure 3)."*

Figure 3 shows the records on the depth scale. How can you conclude from this that the minima and maxima mismatch over time? You would first need to place the cores on the same depth scale by aligning them to a tie point.

> *Answer: We understand that maybe it is not so clear in this figure. But if you look at Figure S2 in the Supplementary Materials you will clearly see the lack of correspondence between the maximums and minimums of the cores. Perhaps adding the timescale to the figure would help to realize that there is no match?*

P 17 L 447f. This belongs into the discussion.

> *Answer: We sorry, but we think it doesn't belong to the discussion. In this case we are just describing results.*

P 17 L 462 ff. Did you assess the possibility that isotopic signals near to stratigraphic features might be altered, i.e., that hoar frost and ice lenses (probably resulting from some melting?) may have modified the isotopes?

> *Answer: Probably partially it is altered. But we think not enough to erase some climatic information, since ice lenses are very thin. Another further study on diffusion would be necessary.*

P 18 L 473f.: Reduce the sentence to "This may indicate that the isotopic signal …"

> *Answer: We changed the sentence to...*

> *"This may indicate that the isotopic signal is maintained throughout the moisture transport in the atmosphere and the snowpack of the Criosfera 1 site (Craig, 1961; Dansgaard, 1964; Masson-Delmotte et al., 2008)."*

Do you mean here "in the snowpack"? Something is missing at the end of the sentence.

> *Answer: Exactly, we sorry by the missing word.*

P 18 L 476: Delete "co-isotopic".

*Answer: Ok. It was deleted.*

Figure 4d and e: I do not really understand what you are showing in these two figures and what the outcome/benefit of/from this analysis is. Why are the correlation coefficients larger than -1/+1? You first state that there is no linear relationship between d excess and δ-values and then you analyse the temporal variability of the linear relationship (i.e., of the slope). Maybe I misunderstand your statistical analysis. Please explain.

> *Answer: There is a misunderstanding here. The correlations are given by blue line, and they range between −1 and +1. Exactly. If we plot the individual results, there was no linear relation. Obviously, this is because temporally such a relationship is not stable.*

I also do not understand what you mean with your last sentence, i.e., that the changes in the relations are not justified by the stratigraphy.

> *Answer: It means that such changes are not justified by stratigraphy features as ice lenses, depth hoars, etc.*

P 18 L 484 ff. You are listing here statistically not significant trends. P-values of 0.75 and 0.87, which strongly exceed your α-level, mean that were is no real trend in your data. The same is valid for the d excess with p-value 0.2. Hence, stating trends here in ‰/year is misleading, as they are most likely to occur by chance, at least this is what the high p-values imply. Just say that no statistically significant trends were found, either in the δ-values or in the d excess. The only trend that can be considered statistically significant is the d excess trend in TT01 for 2000-2011 with p-value 0.06. Leave out the numbers for all other "trends".

> *Answer: Ok. We understand your point. But we stated that there was no significant trend. Please, see below (highlighted in red).*
>
> *"The trends of the isotope compositions time series of both firn cores (TT01 and CR1) were analysed. TT01 firn core presented non-significant isotopic trends […] The CR1 core also presented non-significant isotopic trends, […]"*
>
> *However, we agreed to reduce the paragraph's excessive information. See below our new proposal:*
>
> *"The trends of the isotope compositions time series of both firn cores (TT01 and CR1) were analysed. TT01 firn core presented non-significant isotopic trends from 1999 to 2014. The CR1 core also presented non-significant isotopic trends, but for the period 2000-2011. However, for the same period of CR1, the d-excessTT01 trend is increasing significantly, with slope of 0.11 ‰ yr-1 (p > 0.05 (0.06); α = 0.05)."*

P 21 L 506ff. The same as before: Do not list slopes as trends which are statistically not significant. Only the d excess composite record shows are statistically significant trend and therefore has to be included with its slope and p-value.

> *Answer:* *Ok. We changed this paragraph to...*
>
> *"Composite records were constructed for each isotopic parameter in order to better evaluate the isotopic trends in Criosfera 1 site (Figure 5). As shown in Figure 5, these records cover the period from 2000 to 2011. For this period, none statistically significant isotopic trend was observed for δs. On the other hand, a statistically significant trend is verified in d-excesscomposite record (slope of 0.09‰ yr-1 (p < 0.05 (0.04); α = 0.05))."*

Figure 5: The trend lines in (a) and (b) are misleading since the "trends" are not statistically significant. Remove them. In general, I would recommend to only show the d excess with its statistically significant trend. Please provide either r or R2, otherwise you provide the same information twice.

Figure 6 and Figure 8: Please include the p-values in the figure. Then you can refer to Figures 6 and 8, whenever you have statistically not significant correlations without mentioning all the values in the text.

> *Answer:* *Our sincere apologies RC1, but we don't understand why in one figure we must remove non-significant data and in others we must list them.*
>
> *We think that if we add the p-values to figures 6 and 8 they will be very polluted. So, we think it's best to leave only the significant correlations highlighted in bold. I.e., it is better to mention some in the text than to overload the figure with more data. About Figure 5, maybe we can remove the R2 values.*

P 22 L 524: Please rephrase the last part of the sentence. I guess you want to express that the composite record shows the same relationships as TT01.

> *Answer:* *We changed the las phrase to...*
>
> *"Further, we evaluated that the isotopic records of the TT01 would be more appropriate to perform climatic correlations due to the nosier signal of the CR1 core (section 3.2) and because the composite record shows similar relationships as TT01 core (Table S2)."*

P 22 L 530 ff. Again, you provide values for statistically not significant correlations. If correlations with autumn and summer SAM are not statistically significant it also makes no sense to calculate running correlation for these two periods.

> *Answer:* *We excluded all the values related to non-significant correlations as...*

*[…] and non-significant weak correlation with autumn and summer SAM indices. […]*

*About the running correlations, we chose to mention to make it clear that not all correlation with SAM were stable over the time.*

P 22 L 537f. This sentence belongs to the discussion.

> **Answer:** *Our sincere apologies. Nevertheless, we think it does not belong to the discussion. Here we are just describing a result.*

P 22 L 551 ff. The same as above. The correlations of the d excess with WSSSIC and ABSSSIC are not statistically significant. Hence, as there does not exist any relationship between these parameters, it makes no sense to analyse them for their temporal stability.

> **Answer: Sure.** *We changed the paragraph to...*
>
> *"The d-excess correlated moderately with the latitude of ASL (significant and positive correlation; r = 0.5; p < 0.05) (Figure 6). The d-excessTT01-ASLlatitude relationship was positive over the 2002-2014 period (Figure S4e) and stable from 2005 onward. "*
>
> *We have considered deleting Figure S4f.*

P 23 L 560: The correlations were verified by what? Something is missing here.

> **Answer:** *We changed the sentence to...*
>
> *"No significant correlation was found between standardized $\delta^{18}O$ and the number of HSD and the annual hourly frequency of SWE."*

P 23 L 560 ff. The same as above, all presented correlations are not statistically significant. So just state that you did not find any statistically significant correlations instead of listing all the values which are of no importance. Also, if a relationship is statistically not significant, you cannot call it a "weak relation", because there is just no real relationship.

> **Answer:** *Ok. We changed the paragraph to...*
>
> *"To verify if interannual variations in the frequency of EPE and SWE explain the stable water isotope compositions variability at the Criosfera 1 site we computed correlations between these parameters both for records of TT01 and CR1cores (not shown in Figure 6). No significant*

*correlation was found between standardized δ18O and the number of HSD and the annual hourly frequency of SWE."*

Figure 6: Please explain in the captions what are aSW and aHSDW.

> *Answer:  We added the explanation of these abbreviations in the caption of Figure 6.*

P 24 Chapter 3.5 Please round all accumulation rates to a maximum of two decimal places.

> *Answer:  All accumulation rates values were rounded.*

Again, you provide numbers for trends that are statistically not significant. Just say that accumulation rates show no statistically significant trends.

> *Answer:  Ok. We excluded these numbers. The new paragraph is given below:*
>
> *"The mean annual snow accumulation rates derived from the TT01 core are similar for both the 1999-2014 and 2000-2011 (both 0.24 w. eq. yr-1) periods. The average snow accumulation was 0.24 ± 0.07 m w. eq. and the TT01snow accumulation time series exhibited a non-significant slight decreasing trend. Contrastingly, the mean snow accumulation rate derived from the CR1 core was 0.28 ± 0.01 m w. eq. yr-1 for the 2000-2012 period. The average snow accumulation is 0.32 ± 0.13 m w. eq. and the CR1snow accumulation time series exhibited a non-significant slight increasing trend. Table 5 lists the snow accumulation rates (m w. eq. y-1) for different periods computed of both cores (TT01 and CR1). Noticeably, the snow accumulation rates of the TT01 core had opposite behaviour in relation to the rates of the CR1 core, considering the three-time intervals analysed: 2000-2003, 2004-2008, and 2009-2012."*

P 24 L 580: 2012-2000 period?

> *Answer:  It was corrected.*

P 24 L 584 and Table 5: What is the reason behind choosing these three time periods for calculating accumulation rates?

> *Answer:  First, these are the periods that best allow for observing this inverse behavior between the accumulation rates of the two cores used in our study. And second, the time span of these periods is almost the same.*

Time periods in the Table have to be reversed, e.g., 2009-2012 instead of 2012-2009.

> **Answer:** It was corrected.

P 24 L 590 f.: This belongs to the methods section.

> **Answer:** We added this information in the section 2.4 (Dating of firn cores and estimation of annual snow accumulation rates) as...
>
> "After dating both cores, annual snow accumulation rates were determined and presented in meters of water equivalent per year (m w. eq. y-1). For this, the real depth was previously converted in m w. eq. using the densities of the core sections calculated during the fieldwork (section 2.1). We obtained a local average annual snow accumulation stacking the AWS and firn core data. For this, the snow height measurement obtained by the ultrasonic sensor installed near the Criosfera 1 AWS was multiplied by the average surface density (i.e., by 0.37 g cm-3; see below section 3.2.)."
>
> In the section 3.5 (Annual snow accumulation) we only mentioned the value of the local average as...
>
> "We obtained a local average annual snow accumulation of 0.24 ± 0.09 m eq. w. per year over 1999-2018. […]"

P 24 L 598 f.: You have to keep in mind that the ERA5 data is calculated by a model for certain grid points that are usually at another altitude than your actual study site. What is the altitude of the used ERA5 grid point(s)? Please add the ERA5 grid point(s) used in Figure 1. ERA5 does not capture orographic effects on precipitation very well, hence it is no surprise that trends in ERA5 do not match trends in firn core data. Actually, it is quite surprising that both records match quite well in some periods.

> **Answer:** Sorry, but what do you mean with the altitude of the used ERA5 grid point(s)? The grid point (-84 (lat); -79.5 (long)) is coincidentally the same as the Cryosphere 1 site (~-84°S, ~79.5 °W) (as mentioned in the section 2.5). Therefore, we thought that is not necessary add it in Figure 1.

Figure 7: Delete "the" before ERA5.

> **Answer:** Ok. "The" was deleted.

Was the snow accumulation composite record derived by averaging or linking? This is not clear.

*Answer: Both. First, we standardized the accumulation series obtained from both the cores (TT01 and CR1) and the Criosfera 1 AWS using the local mean and standard deviation. Then, we took the averages between the correlated periods of the cores (2000-2011). Finally, we linked the standardized accumulation series of the non-correlated period (2012-2018) with the average's series obtained for the correlated period (2000-2011).*

P 25 L 623 to P 26 L 650: These paragraphs belong to the discussion.

*Answer: Sorry, we disagree with you. We are just describing results which are in Figures 9, S9-S13.*

P25 L 603f. If the p-value largely exceeds 0.05 then the correlation between ASL and WSS pressures is not statistically significant and hence cannot be further interpreted.

P25 L 607 It is not necessary to provide the correlation values for the SOI index as they are not statistically significant.

*Answer: We changed the paragraph to...*

*"The observed annual snow accumulation over 1999-2018 related positively and moderately with annual SOI index (r = 0.60; p < 0.05 and α = 0.05), concentration of sea ice in the WSS (r = 0.51; p < 0.05 and α = 0.05), and weakly with mean pressure of the ASL and WSS (both not statistically significant) (Figure 8). Further, it correlated negatively and moderately with snowfall non-weighted mean temperature (not statistically significant) (Figure 8). Significant, positive, and moderate correlations with summer (r = 0.58; p < 0.05 and α = 0.05), autumn (r = 0.44; p = 0.05 and α = 0.05) and spring SOI indices (r = 0.50; p < 0.05 and α = 0.05) and no relation with winter SOI index were also verified (not shown in Figure 8). [...]*

P 25 L608 ff. If the running correlations are statistically significant, the p-value range should also be provided.

*Answer: Sure. we thought that it is not appropriate to put it because we would be putting the p values only for the extremes. However, we can insert in the text.*

P25 L 614 Delete the statistical values for the not significant correlation with the SAM.

*Answer: These values were deleted.*

P 25 L618f. This sentence is misleading. The relationship is not statistically significant, so it is wrong to state that the HSD frequency is related to ASL latitude.

P 25 L 625 ff. This sentence is confusing. Did the accumulation also start in spring and late winter in 2013 and 2015? Please rephrase.

*Answer:* *Ok. We changed the sentence to...*

*"As shown in Figure 9a and Figure S9, the EPE occurred more concentrated in the late winter and spring months and the preserved accumulation started, respectively, in these periods in 2013 (Figure 9) and 2015 years (Figure S10)."*

Delete "years" after 2015.

*Answer:* *"years" was deleted.*

P 26 L 640 ff. Does the value 0.06 m snow height result from ERA5 or in-situ measurements? Did you check how well ERA5 reproduces snowfall and hence accumulation at your site, since ERA5 generally tends to underestimate precipitation amounts, at least in West Antarctica (see Tetzner et al., 2019, doi:10.3390/geosciences9070289). Otherwise, it is a bit difficult to draw reliable conclusions from the differences in ERA5 snow height and snow height ultrasonic sensor measurements.

*Answer: Of course, we know the ERA 5 tends to underestimate the precipitation. As we have tried to explain in this paragraph clearly this underestimation is related to inability of ERA5 to computed additional deposition and erosion caused by blowing snow and snowdrift events – common processes that occur at the studied site.*

P 26 L 645f. Rephrase to: "...dominantly from the SE sector and rarely exceed 15.5 m/s at 850 hPa" (round up to a maximum of one decimal place).

*Answer:* *We changed the sentence to...*

*"Fifth, events that drive high snowfall rates at the Criosfera 1 site weaken the south wind, come dominantly from the SE sector and rarely exceed the 15.51 m/s at 850 hPa (Figures 9, S9-S11, and S12)."*

*We think that in this case, it is not necessary to round, after all this is our threshold for SWE and we present all data so far to two decimal places.*

P 26 L 648: Rephrase to: "air-mass incursions from the Peninsula region to the east coast of the WSS have intensified ...".  Or what do you mean with "air-ass incursions by WSS"?

*Answer:* We changed the sentence to...

*"[…] air-masses incursions from the Peninsula region to the east coast of the WSS have intensified in the first two decades of the 21st century compared to the last century (Figure S13) […]"*

P 26 L 647ff. Do you have any further evidence, i.e., references for the stated trends in air mass incursions and westerly winds or do you simply conclude this from your own calculations? I think some references are needed here.

*Answer:* Our own calculations using ERA5 datasets supported this affirmation as shown in Figure S13. We used the Climate Reanalyzer (UMAINE) to calculate these anomalies.

**Discussion**

In general, the discussion section definitely has to be divided into several sub-chapters in order to better guide the reader through your thoughts. I also would be more careful with absolute statements. Your data suggests certain relationships between isotopes and influencing climate factors, but you cannot be sure by a hundred percent.

*Answer:* Certainly, we understand what you mean. Our idea was to have an integrated discussion to further highlight the most important findings, discussing them in descending order of importance. Sorry if it sounded like we discussed something with 100% certainty. It was not our idea, so much so that we stated that further studies with new cores would be necessary to check whether such relationships will persist.

P30 L 674-678: In my view, this paragraph belongs to the conclusions.

*Answer:* Here our intentions was only remembering what we did and our aims before starting the discussions.

P 30 L 680-687: In my view, this belongs to the introduction. There you should explain the role of the SAM for Antarctica.

*Answer:* Ok. We can transfer such information to the introduction, but we are afraid of it becoming a lost paragraph, that is, not fitting into the line of reasoning that we thought for the introduction.

P 30 L695ff.: The same as above. I think the explanation of the ASL should go into the introduction.

*Answer:* *The same as answers to question about "P 30 L 680-687".*

P 30 L 698f. But your study site is rather located in East Antarctica or at least at the intersection between Antarctic Peninsula, WAIS and EAIS. How do you know that warm air masses reaching the WAIS and the Antarctic Peninsula during a positive SAM phase and deep ASL also reach your study site? Or maybe I get you wrong here?

*Answer:* *We sorry, but perhaps you are wrong. In fact, our study area is located in the intersection between WAIS and EAIS in the WSS. However, it is already known that EAIS it's not so impenetrable. Furthermore, as we mentioned throughout the discussions, recent studies have shown the relatively warmer air masses which can reach the South Pole (e.g., Clem et al. 2020)*

*References: Clem, K. R., Fogt, R. L., Turner, J., Lintner, B. R., Marshall, G. J., Miller, J. R., & Renwick, J. A. 2020. Record warming at the South Pole during the past three decades. Nature Climate Change, 10(8): 762-770.*

You showed earlier that during days with (high) snowfall events, which probably produce your isotopic signal (HSDs – 42% of annual precipitation), a S/SE-wind regime prevails. Also, SWEs which may contribute to your isotopic signal through removal and redistribution of snow are connected to southern winds. Warm air masses advected towards the WAIS and Antarctic Peninsula during a positive SAM phase predominantly come from the west/northwest. So, it seems contradicting to me to connect the negative/positive correlation between ASL pressure/SAM and your isotopes with the warm air advection to the WAIS and Antarctic Peninsula (unless the warm air masses cross the mountain range of the Antarctic Peninsula to reach the Weddell Sea sector). Of course, you have the negative correlation between the isotopes and the ASL pressure, which I guess partly results from the fact that your isotopes are positively correlated with the SAM and that the SAM in turn is negatively correlated with the ASL pressure. I think backward trajectory modelling would be essential to better understand the relationship between your isotopes and air mass advection from different directions.

*Answer:* *We agreed that a backward trajectory modelling could be interesting. But we have field evidences that relatively warmer air masses that reach Criosfera 1 site come from the east site of Antarctica Peninsula (personal communication: Professor Francisco Eliseu Aquino). And simulated data of previously studies of our group (e.g., Marcher et al. 2022) has also shown this direction and pointed out that some warmer air-masses can overcome the orographic barrier in the intersection between Peninsula and WAIS.*

P 30 L 700: You should introduce the IPO already in the introduction section.

*Answer:* *We have not correlated any of our data with the IPO, therefore we think it is not necessary to mention it in the Introduction.*

P 32 L 755ff. The correlations you are discussing here are not statistically significant.

> **Answer:** Ok. But we make clear, e.g., "correlations with sea ice in the ABSS and WSS hardly explain an increase in d-excess at the Criosfera 1 site..."
>
> Concerning the d-excess/δs slopes and d-excess-δs, we were referring to Figures 4d and 4e.

**Technical corrections**

P1 L17: Rephrase to: "Therefore we investigated the relationship between isotopic (..) and snow accumulation data […] and glaciological information, […]"

> **Answer:** We changed the sentence to…
>
> "Therefore, we investigated the relationship between ($\delta^{18}O$, $\delta D$, and d-excess) and snow accumulation data of two shallow firn cores (both ~9.0 m deep) and glaciological information, local and regional meteorological data (both ERA5 and AWS), indices of large-scale atmospheric modes (as SAM and ENSO) and the Amundsen Sea Low (ASL)".

P 3 L72: Rephrase to: "[…] that occurs between evaporation from the moisture source and deposition at a specific site […]"

> **Answer:** We changed the sentence to…
>
> "Nonetheless, as the isotopic compositions imprint each that occurs between evaporation from the moisture source and deposition at a specific site and afterwards, they also store a range of other information that may either obliterate or change δs-temperature relationships on shorter timescales in a given area (Jouzel, 2013; Touzeau et al., 2016; Landais et al., 2017)."

P3 L 76: "on local and regional scales"

> **Answer:** We changed the sentence to… "[…] (on local and regional scales) […]"

P3 L82: Use plural here: "… changes in moisture sources…"

> **Answer:** We correct the plural. The sentence changed to…

*"[…] d-excess (a secondary isotopic parameter sensitive to kinetic effects; calculated by: $d = \delta D - 8 \times \delta^{18}O$; Dansgaard, 1964) is used to explore changes in moisture sources and atmospheric paths."*

P4 L101: "… circulation on large and synoptic scales"

*Answer: The sentence changed to… "[…] and the atmospheric circulation on large and synoptic scales […]"*

P4 L108: "Antarctic ice-core studies …"

*Answer: The sentence changed to… "Further, although Antarctic ice-core studies have reaffirmed that [...]"*

P4 L114: "… in each Antarctic sector"

*Answer: The sentence changed to… "[…] and accumulation records on a basin-scale in each Antarctic sector […]."*

P4 L116: Delete "the" before "two high-resolution shallow firn cores"

*Answer: Ok… "the" was deleted as recommended. The sentence changed to…*

*"Here, we investigate the stable isotopic content ($\delta^{18}O$, $\delta D$, and d-excess) and the snow accumulation variability from two high-resolution shallow firn cores drilled on the upper reaches of the Möller Ice Stream basin, Weddell Sea Sector, in the transition between West Antarctica Ice Sheet (WAIS) and East Antarctica Ice Sheet (EAIS)."*

P4 L117: "…between the West Antarctic Ice Sheet and the East Antarctic Ice Sheet …"

*Answer: Ok. We changed the sentence to…*

*"[…] in the transition between the West Antarctic Ice Sheet (WAIS) and the East Antarctic Ice Sheet (EAIS)."*

P4 L 119: Use plural here: "post-depositional processes"

*Answer:* Ok. We correct the plural. The sentence changed to…

*"[…] (i.e., explore the annual accumulation pattern, probable events preserved, and post-depositional processes) […]"*

P4 L 125 f.: Delete "high snowfall days" and use only the previously introduced abbreviation.

Rephrase to: "… in order to assess from which sector the storms that reach the study site come." (I guess this is what you wanted to say, didn't you?)

*Answer:* Ok. But we deleted this sentence to reduce information about the methodology described in the introduction as requested.

P4 L 128: I think the abbreviations ABSS and WSS have not yet been explained. Explanation in the abstract does not count here.

*Answer:* We changed the sentence to…

*"We explored the relationships between annually averaged snow accumulation and the number of HSD, frequency of SWE, the temperature at 900hPa (weighted and non-weight with precipitation), sea ice concentration (SIC) both from Amundsen Bellingshausen Sea sector (ABSS) and Weddell Sea sector (WSS), large-scale atmospheric modes (as Southern Annular Mode (SAM) and El Niño-Southern Oscillation (ENSO)), and pressure in both Amundsen Sea Low (ASL) and WSS."*

P5 L 143: "The Criosfera 1 site is located in the MIS basin catchment area …."

*Answer:* Ok. We changed the sentence to…

*"The Criosfera 1 site is located in the MIS basin catchment area ($\sim$84°S, $\sim$79° 30'W; at $\sim$1276 m above sea level (a.s.l.)), near the boundary with Foundation Ice Stream Basin (FIS basin), and approximately 650 km of South Pole (Figure 1a)."*

P5 L 145: Delete the hyphen.

*Answer:* Ok.

*"These basins lie on the Ronne Embayment region which is drained by glaciers and ice streams that feed into the Filchner-Ronne Ice Shelf (Figure 1a)."*

Caption Table 1: "… temperature at 10 m depth"

*Answer: Ok.*

*"Table 1. Details on two firn cores from the Criosfera 1 site used in this study and other information: location, date of drilling, the period covered and depth of the cores, number of the samples used in this work, sample resolution, mean density and borehole temperature at 10 m depth."*

Caption Figure 1: "… glacier outlines at 250 m depth"

*Answer: Ok.*

*"Figure 1. […] The thin blue dashed lines are ice streams and glacier outlines at 250 m depth. […]"*

P 6 L 171: "The Criosfera 1 site …"

*Answer: Ok. We changed the sentence to…*

*"The Criosfera 1 site is a region characterized by strong temperature contrasts."*

P 7 L 176: Use plural here: "… cold conditions … are broken off…"

*Answer: We correct the plural. The sentence changed to…*

*"However, generally, the extremely cold conditions in both late autumn, winter, and early spring are broken off by incursions of relatively warmer air masses coming from the ocean (Nicolas & Bromwich, 2011)."*

P 7 L 180f.: "low-scale dune fields of 20-40 m height"
Delete "in wide extension".

*Answer: We changed the sentence to…*

*"Due to the extreme wind regime, the presence of sastrugi and low-scale dune fields of 20-40 m height is well marked around the study area (Marcher et al., 2022)."*

P 7 Title chapter 2.2: Firn/ice core collection

*Answer: Ok. **Changed to…***

*"2.2. Firn/ice core collection"*

P 7 L 191: Delete "at".

    *Answer:* *Ok. "The preposition "at" was deleted".*

P 8 L 224: Add "a" before "class 5 clean room".

    *Answer:* *Ok. The article was added. The sentence changed to…*

    *"This system is installed in a class 5 clean room with high efficiency particulate air (HEPA) filters."*

P 9 L 245: "…the analytical method used for each core…"

    *Answer:* *Ok. We changed the sentence to…*

    *"Table 2 summarizes the analytical method used for each core, the chemical parameters, and the number of samples analysed."*

P12 L 343: Insert "it" before "overestimates the temperature and …"

    *Answer:* *Ok. We inserted it in the sentence as…*

    *"In addition, the ERA5 satisfactorily reproduces the variability of the meteorological parameters at the Criosfera 1 site, although it overestimates the temperature and underestimates the wind velocity (see Table S1)."*

P13 L 376: Delete the colon after "coastal areas".

    *Answer:* *We changed the sentence to…*

    *"Our threshold for SWEs is below of that considered in coastal areas, i.e., from gale velocities (> 17.2 m/s), according to the Beaufort Wind Force scale (Turner et al., 2009b)."*

Title of 2.8: Replace the first "and" by a comma.

    *Answer:* *Ok. The new title is …*

*"Time series construction, trend, and correlation analysis"*

P 15 L 425: "covers 16 years"

*Answer:  We corrected the verb conjugation. The new sentence is...*

*"In total, the TT01 firn core covers 16 years (from 1999-2015), with an estimated uncertainty of ± 0.41 (< 5 months)."*

P 17 L 449: "… has a noisier signal…"

*Answer:  We corrected the sentence to...*

*"Remarkably, the CR1 has a noisier signal than TT01."*

P 17 L 456: Delete "cores" after CR1.

*Answer:  It was deleted.*

P 17 L 459: Add a comma before "then".

*Answer:  Ok. Comma was added before "then".*

P 17 L 464: Replace the colon by a comma.

*Answer:  Ok. Colon was exchanged by a comma.*

P 18 L 479: Replace "unjustified by "not justified".

*Answer:  Ok. We exchanged "unjustified" by 'not justified' as...*

*"Changes in these two relations are not justified by stratigraphic characteristics (e.g., ice lenses and depth hoar)."*

P 21 L 506: Replace "to" by "for".

*Answer:* We exchanged "to' by 'for" as...

"For this period, none statistically significant isotopic trend was observed for δs: …"

P 24 L 590: "firn core data" (delete the "s")

*Answer:* Ok. We deleted the "s".

P 24 L 590: Add "the" before "composite record".

*Answer:* "The" was inserted before "composite record".

P 25 L 620: Delete "the" before "high snowfall rate".

*Answer:* Ok. It was deleted.

P 25 L 631: Rephrase to: "Furthermore, in 2016 there was no additional accumulation towards the end of the year, as most HSDs took place in the middle of the year."

*Answer:* Sure. We have reformulated the sentence as suggested.

P 26 L 634: Replace "in" by "at the".

*Answer:* Ok, "at the" replaced the "in".

P 26 L 644 and L 645: Add "the" before "Criosfera 1 site".

*Answer:* Ok. "The" has been added before "Criosfera 1 site".

P 26 L 647: "Data on the mean …"

*Answer:* We changed the sentence to...

*"Data on the mean annual meridional wind anomaly show that air-masses incursions by WSS, from the Peninsula region to the east coast of the WSS, have intensified in the first two decades of the 21st century compared to the last century (Figure S13), and indicate that moisture comes primarily from this sector."*

P 30 L 688: Change "trigger" to "triggers".

> ***Answer:*** *We corrected the verb conjugation as...*
>
> *"[...] the SAM (+) triggers an increased upward moisture [...]*

P 30 L 695: Rephrase to: "...the SAM strongly influences the depth of the ASL..."

> ***Answer:*** *We rephrased the sentence as suggested.*

P 30 L 700: Rephrase to: "... are linked to the shift of the Interdecadal Pacific Oscillation to the negative phase".

> ***Answer:*** *We rephrased the sentence as suggested.*

P 31 L 716: Delete "For".

> ***Answer:*** *Ok, "for" was deleted.*

P 33 L 797f.: Add "the" before "EPE signal", delete "the" before 1999. Use "approaching" instead of "has approached".

> ***Answer:*** *We corrected all these minor mistakes. The new sentences are shown below:*
>
> *"We extend the interpretation that mainly the EPE signal is preserved until 1999 due to seasonal bias of the isotopic records from the Criosfera 1 site. An explanation for accumulation 2015-2018 approaching even closer to snowfall than in the 2002-2005 period is that in the former period, there was an intensification of negative sea ice anomalies in the WSS linked to depressions increase in WSS (Turner et al. 2020), which could have interfered in the intermittence of the EPEs."*

P 34 L 812: Delete one "also".

> ***Answer:*** *Ok. We deleted the second "also".*

P34 L 820: "The stratigraphic analysis…"

>**Answer:** *Ok, we corrected this minor mistake.*

P 34 L 834: "… ice lenses were observed…"

>**Answer:** *Ok. We corrected the passive voice use.*

P 35 L 841: "Since spatial variations of accumulation …"

>**Answer:** *Ok. We corrected this minor mistake.*

P 34 L 843: Use "are located" instead of "lied".

>**Answer:** *We exchanged "lied" by "are located" as suggested.*